# The structure and statistics of language jointly shape cross-frequency neural dynamics during spoken language comprehension

Hugo Weissbart [1,2] ✉ & Andrea E. Martin [1,2]

Humans excel at extracting structurally-determined meaning from speech despite inherent physical variability. This study explores the brain's ability to predict and understand spoken language robustly. It investigates the relationship between structural and statistical language knowledge in brain dynamics, focusing on phase and amplitude modulation. Using syntactic features from constituent hierarchies and surface statistics from a transformer model as predictors of forward encoding models, we reconstructed cross-frequency neural dynamics from MEG data during audiobook listening. Our findings challenge a strict separation of linguistic structure and statistics in the brain, with both aiding neural signal reconstruction. Syntactic features have a more temporally spread impact, and both word entropy and the number of closing syntactic constituents are linked to the phase-amplitude coupling of neural dynamics, implying a role in temporal prediction and cortical oscillation alignment during speech processing. Our results indicate that structured and statistical information jointly shape neural dynamics during spoken language comprehension and suggest an integration process via a cross-frequency coupling mechanism.

Humans comprehend language despite high variability in the physicality of speech acoustics, which can stem from noisy environments or from variations in speakers and their accents. Such robustness in perception may stem from the projection of stored linguistic knowledge via the anticipatory mechanisms of predictive processing. Predictive processing is a widespread neurocomputational principle, or more generally, a framework in which predictions play an active role in the processing of upcoming information streams in the brain[1]. The incoming information may be sensory or solely endogenous neuronal activity from other brain regions, and the output of such processes may be perceptual, motor, or cognitive[2–5]. Although this framework originates from the field of perception[6], evidence of such processing during language comprehension is also abundant[7–12].

Although language comprehension is remarkably adaptable to variation in speech acoustics, the inherent unpredictability of novel messages-each encoding the intended meaning from the speaker-poses a challenge. This unpredictability is not absolute but relates to the relevance and specificity of each message, which often contains sequences and structures that are nevertheless anticipated based on knowledge from previous linguistic experience. There are different levels of granularity, each benefiting from prediction from other representations, such as phonemes predicted from preceding items and spectro-temporal characteristics of the sound segment[13,14]. Furthermore, an essential feature of human language is its display of nested syntactic structures over which meanings are computed[15,16]. Traditionally, language's inherent unboundedness and generative

[1]Donders Centre for Cognitive Neuroimaging, Radboud University, Nijmegen, The Netherlands. [2]Max Planck Institute for Psycholinguistics, Nijmegen, The Netherlands. ✉e-mail: hugo.weissbart@donders.ru.nl

aspect have often been seen as being in putative opposition to distributional and statistical accounts of language processing (e.g., the traditional view on the *competence hypothesis*[17]; or grammar-free account of comprehension[18,19]). In contrast to this dichotomy, in the present study, we synthesise these positions and present a framework wherein the syntactic structure and statistical cues are jointly processed during comprehension[20–22]. We build upon the work of Brennan et al.[23] who demonstrated the sensitivity of the BOLD signal to both structure and surprisal but focused on the localisation of such effect. Moreover, their study, while also investigating naturalistic speech comprehension, could not analyse the temporal dynamics or high frequency activity due to the poor temporal resolution of the BOLD signal. In the present study, we can leverage the use of magnetoencephalography (MEG) to investigate phase and amplitude dynamics of band-limited cortical activity in response to variation in both statistical cues and syntactic features.

To operationalise syntactic processing, we constructed a set of word-level features which describe critical characteristics of the underlying constituency parse or tree structure. Those syntactic features are derived from hierarchical trees operating at the sentence level and are de-lexicalised. They represent aspects of syntactic trees (viz., number of brackets, depth in the tree) which by themselves can elicit a response reflecting tracking of syntactic structures as in Ding et al. and Frank and Yang[24,25]. Similar syntactic metrics have been used to study the effect of syntactic operations such as unification (the "merge" operation in the minimalist program[15,26]) or integration of an item into a larger structure and the depth, a proxy for ongoing complexity, of the syntactic tree at a given word[27,28]. These metrics align with foundational linguistic theories and have been shown to elicit predictable neural responses[16,23,29,30]. Since those features are derived from the constituency parse, we refer to them as rule-based features, in contrast to surface statistical features, which are estimated from sequences of words with no information beyond the sequence itself[25,31]. However, this does not necessarily imply that such statistics will not carry any information about hierarchical structures or about syntax, a topic that is currently under debate[32–34]. What is indisputable, however, is that such statistical metrics are built from a sequence of words and optimised to learn a probability distribution conditioned on the sequence context of preceding words. To learn this distribution, words are presented to a system, such as a recurrent neural network, and the system's task is to predict the next word based on the series of preceding words encountered so far (this memory is embedded in the model's architecture). Non-linear models, like Long Short-Term Memory network or Transformers, can manipulate information such that they could potentially encode structured information within their memory (hidden states, latent representations). From an information-theoretic perspective, surface statistics can give an estimate of the self-information (viz., surprisal) and uncertainty (viz., entropy) measured at the word level while conditioning on the observed context of previous words. Importantly we note that these models, despite their increasingly complex architecture, do not have rule-based knowledge (structurally-)embedded in their learning algorithm and solely make predictions from an estimate of the conditional probability distribution over the vocabulary that is updated through their inner (recurrent) dynamics. However, we note that recent models, such as larger GPTx, appear to perform better on a series of syntactic paradigms[32,33] compared to older architectures that were trained jointly with grammatical rules and structural information along with the word sequence (such as RNNG, Grammatical recurrent neural network; but see refs. 28,35,36). Finally, we note that such statistical cues have been shown to modulate cortical activity during language comprehension[27,37–39].

In light of this apparent dichotomy, and in the context of the debate in cognitive science regarding the role of statistical information in language processing[17,31,40,41], we then ask to what extent do their individual contributions explain neuroimaging data, is the whole better than the sum of its parts? We hypothesise that they jointly contribute to explaining variance in the MEG data while presenting overlapping spatio-temporal sources. Moreover, the dynamics might disentangle them further as predictions and statistical inference seem to be a widespread phenomenon in cortical computation, while the organisation of linguistic units into nested hierarchical structures, at least at first blush, may be related to hierarchical processing in other domains in some ways, but not others[42–44]. We thus further hypothesise that brain responses to statistical and structural features are operated synchronously, with potential distinct time scales, and orchestrated through cross-frequency coupling. Differentiation arises as information flows from one system of prediction to, potentially, another, which would compute different combinatorial aspects of structure building and semantic processing.

To tap into the orchestration of spatio-temporal dynamics through cross-frequency coupling, we measured brain activity with magnetoencephalography (MEG), allowing for a time-resolved recording of neural activity. The temporal resolution of MEG data is fine enough to measure power modulation in a wide range of (high) frequencies together with the phase of a slower frequency range. The role of cortical oscillations in neural computation is still unclear; nonetheless, an increasing number of studies have now attributed some functional role to different frequency bands, not only for low-level perceptual or sensory processing but also in relation to speech processing. Delta and theta-band activity (1-4 Hz and 4–8 Hz respectively) play an important role in the neural tracking of the acoustic envelope[9,45]. It has been hypothesised that low-frequency cortical activity rhythmically modulates neuronal excitability to match the rhythm and landmarks of the acoustic stream, perhaps reflecting speech segmentation mechanism[46–48] but also facilitating the processing of syntactic information via synchronisation to lower level acoustic cues[2,16,49,50]. Moreover, studies have demonstrated how low-frequency neural signals couple their phase with power of higher frequency broadband power (phase-amplitude coupling, PAC thereafter) and such coupling has an impact on behavioural response[51,52]. It has also been proposed that the ongoing phase of neural signals may be modulated by the predictability of words, thus affecting the temporal prediction of syllable and word onset[48,53]. The phase of neural signals is a candidate for carrying information both about statistical and structural features[53–56]. However, it is still widely debated whether measured oscillatory activity in response to speech stimulus truly comes from an oscillatory mechanism or is a by-product of measuring a response to a (pseudo-)rhythmic stimulus, that is, a series of evoked responses. The interpretation of the role, cause and effect, of low-frequency oscillations remains unclear. While some studies claim that there is evidence that neural tracking to speech envelope is solely due to evoked response convolved with acoustic edges[47,57], other studies find evidence consistent with an entrained endogenous oscillator which shows sustained activity and phase-related behavioural modulation[8,10]. We adopt a more agnostic view, focusing on measured phenomena, where we investigate how linguistic features influence neural activity across different frequency bands, by analysing phase consistency, power modulation and PAC. While we prefer to refer to low- and high-frequency activity specifically, we may use the term *cortical oscillations* to refer to band-limited power elevation observed in the MEG power spectra without making any claim about the underlying mechanism.

Our hypothesis draws on the intersection of syntactic processing and predictive coding theories, positing that the brain's response to language is not just reactive but anticipatory, integrating both structural and statistical cues in real-time. We thus propose to link properties of syntactic structures, jointly with information-theoretic metrics to MEG data. Brennan & Pylkkänen[58] and Nelson et al.[27] have used a similar approach to link syntactic features to

electrophysiological data. However, the former studies focused on the localisation of MEG activity to study word-evoked responses, while the latter analysed high gamma activity recorded from intracranial electrodes. In another study, Brennan & Hale[20] used information theoretic metrics built from context-free grammar parsers and delexicalised n-grams, which do not capture semantic information (thus their surprisal metric greatly differs from ours). Finally, in Brennan et al. (2016)[23], a link is made between hierarchical syntactic features (node count) and surprisal from Markov models (n-grams, lexicalised and unlexicalised). While they elegantly show how different parsing strategies affect the prediction of neural activity, the statistics of word predictions are overlooked. Modern autoregressive language models produce a more precise estimate of conditional probabilities, from which we extract surprisal and also entropy, thus allowing for a better account of predictive mechanisms. To our knowledge, the current literature has not explored the interplay of such features across frequency bands with MEG.

Finally, we investigate the relation of word level computation at different timescales, following the work from Donhauser & Baillet[59]. Indeed, they observed distinct roles for theta and delta rhythms during the prediction of phoneme sequences. The authors intepreted effects in theta as a read-out of the sensory sampling mechanism bound to maximize the expected information gain, and their delta effect as encoding non-redundant, i.e. novel, information deviating from internally generated predictions, which results in an update of the internal model[60]. Top-down predictions, but also updates, have been already linked to beta power modulation[61]. All in all, this mechanism, if supported via delta-phase, is bound to endogenously generated predictions and thus to the internal model of the listener. Thus, in line with Donhauser & Baillet[59], we expect a stronger delta-beta coupling for non-redundant information, which is approximated by the *surprisal* feature of the statistical features.

In the present study, we ask whether syntactic and statistical word-level features provide complementary cues to the processing of speech input by investigating how putative cortical oscillations and broadband activity are distinctively modulated by linguistic features. The aim is to discuss how different frequency bands orchestrate the processing of linguistic units. Those units are favoured differently, whether from predictions on sequential statistics, a potentially domain-general feat, or mirror integration into larger linguistic compounds, thus reflecting language-specialised computations. Specifically, we examine how syntactic and statistical features influence phase consistency, power, and phase-amplitude coupling. The Results section will detail our findings on how the brain responds to these linguistic cues, highlighting the separate and combined effects of structural and statistical information on neural processing. Through our analyses, we aim to shed light on the mechanisms underlying predictive coding in language comprehension, offering insights into the neural basis of language processing.

## Results

In order to measure time-resolved cross-frequency coupling in relationship to different linguistic features we first analyse both the presence of word-related phase and power modulation as well as how speech representations perform in predicting MEG signals. First, we extracted linguistic features reflecting both syntactic complexity and statistical properties of speech from the presented stimuli. So called rule-based features are derived from constituency tree, while statistical features are generated using the probability distribution of next word predictions from a large language model. These features include syntactic depth and the number of closing brackets to represent structural properties of syntax[29,58], and word surprisal and entropy for statistics of word-level predictions[27,38]. We construct Temporal Response Function (TRF) models to predict MEG signals based on these linguistic features, enabling a precise examination of how each feature

influences neural responses over time. The analysis pipeline (detailed in the Methods section) involves aligning the MEG data with our linguistic features, then using ridge regression to estimate the TRF coefficients, and finally evaluating the models' performance through correlation analysis between predicted and observed neural activities. In most cases, model comparison is carried out between the true model and a null model for which values (but not timing) of a given feature (or of a feature set) are shuffled. Through this method, we aim to reveal the mechanisms by which the brain integrates and processes linguistic information at different levels. The general analysis pipeline, along with stimulus representations and analysis methods, are presented in the diagram of Fig. 1.

We measured power spectral density (PSD) of the MEG data in order to assess the quality of the data and the presence of neural oscillations. As seen in Fig. 2a, b, the MEG data presents neural oscillations in the alpha and beta bands. We found a marginally significant difference in power between the French and Dutch listening conditions in the beta band (FDR corrected for multiple comparison across all frequencies, corrected $p$-value = 0.06, one sampled $t$-test, dof = 24). We then computed the cerebro-acoustic coherence by taking the magnitude coherence squared between the sensor-level MEG signals and the sound envelope of the stimuli, see Fig. 2c. This reveals coherent phase alignment between sound amplitude and MEG signals in the delta and theta bands. Importantly, while those reflect processing of speech-like sounds they do not directly reflect comprehension as it occurs for both French and Dutch listening conditions. Besides, it is actually significantly greater for the French condition in the delta and theta-band ($p$ = 0.01, cluster-based permutation test using one sample $t$-test as statistic). Finally, we computed a time-frequency representation for every word-epochs to analyse phase consistency and power modulation at higher frequencies. We extracted power modulation and inter-word phase clustering from the complex Fourier coefficients. As seen in Fig. 2d, e, we observe power modulation in the beta band, and inter-word phase clustering in the delta and theta band.

### Phase consistency and power modulation at word onsets

Recent studies have demonstrated the important role of low-frequency cortical activity during speech processing. Whether it is a form of neural entrainment to the acoustic envelope or a series of evoked responses to acoustic edges is debated, but results come together in that they show a stronger correlation between speech spectro-temporal features and MEG or EEG signals in both the delta and theta band[7,45,54]. Etard & Reichenbach[62], for instance, concluded from their study the existence of a dissociable account for each frequency band with respect to the clarity and intelligibility of the speech signal. Modelling work from Hyafil et al.[48] reinforces the idea of a phase alignment within the theta band to promote segmentation of syllabic sequences. Altogether, we hypothesised change in phase consistency in delta and theta range after word onset, potentially coupled with modulation of beta oscillatory activity and with gamma broadband activity (which has already been suggested[51,52,56]). To further validate this choice of frequency bands of interest, we computed word-triggered phase consistency and power modulation. Fig. 2 presents the long-term power spectral density across all stories (panel a) as well as power modulation and inter-trial phase consistency happening after word onsets (panels d,e). We found significant power modulation in the beta bands and significant inter-word phase consistency in the delta and theta bands (cluster-based permutation using one sample $t$-tests as cluster statistics after baseline removal, tested against a null hypothesis of zero-valued population mean). One challenging aspect in computing the inter-trial phase consistency (ITPC, or inter-event phase clustering and in the case of the present study: inter-word phase clustering) is that it relies on the clustering of phase, evidently, across trials (defined by word onset timings). Hence, it depends on trial-based experimental design. We propose an innovative approach to

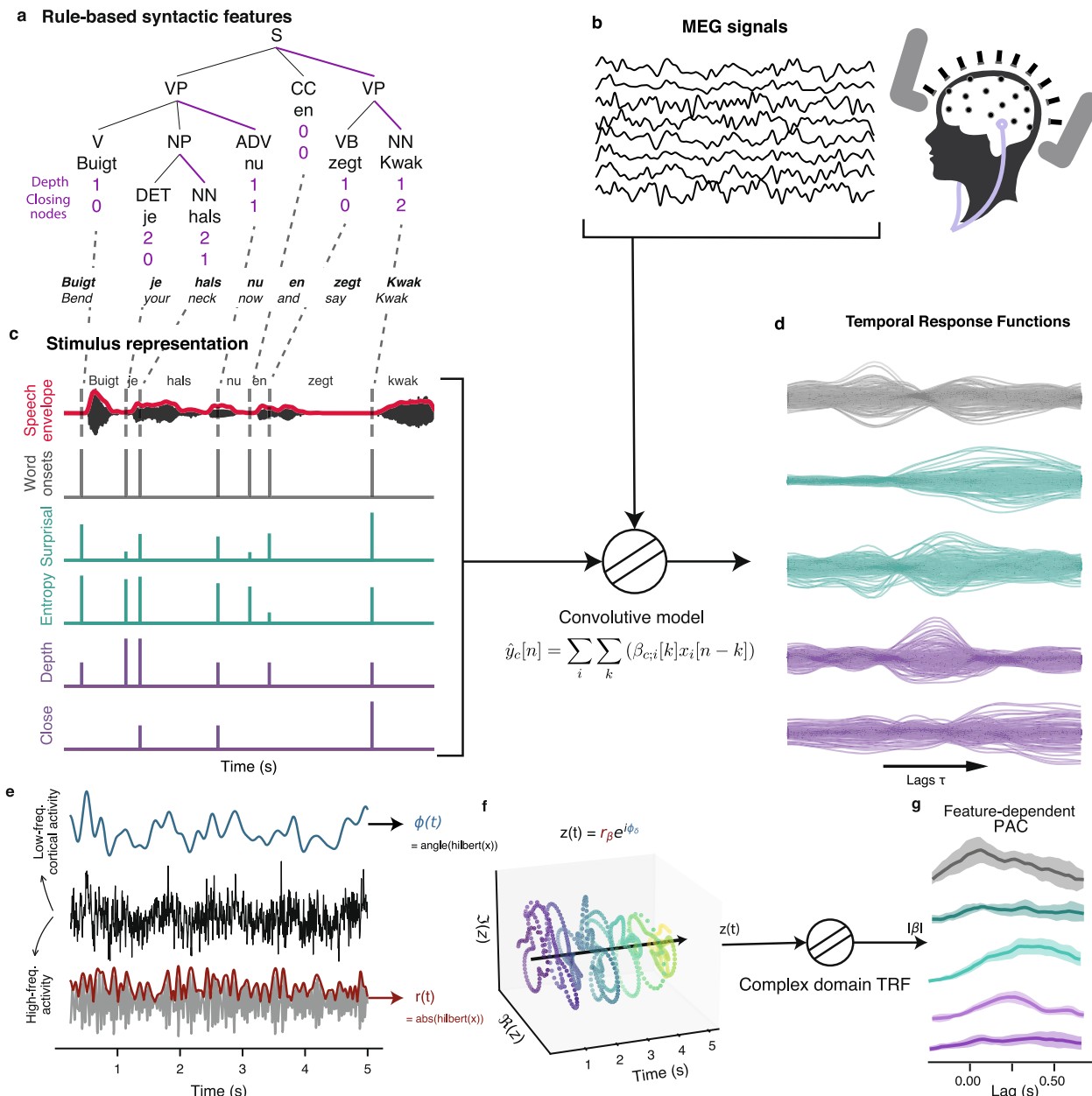

**Fig. 1 | Data analysis workflow and presentation of feature-dependent phase-amplitude coupling (PAC) computation.** An example sentence is shown, along with the syntactic features derived from the constituency tree (**a**), i.e. tree depth and the number of closing nodes, which are shown below each tree leaf (S: sentence, VP: Verbal phrase, NP: Noun phrase, CC: Coordinating conjunction, ADV: Adverb, VB: Verb, NN: Singular noun, DET: Determiner). The MEG recordings (**b**) together with the full stimulus representation (**c** consisting of sound envelope, word onsets, statistical features shown in green and syntactic features in purple) are used to compute the temporal response functions (**d**). Finally, the bottom diagrams (**e**–**g**) summarize how feature-dependent PAC is estimated. **e** We first extract the phase at low-frequency and the amplitude of high-frequency band of the signal. **f** These time series are then combined to form the complex analytical signal. The Temporal Response Function (TRF) is thus computed directly in the complex domain from the latter signal and the amplitude of the resulting coefficients $|\beta|$ are taken as the PAC estimate (**g**).

circumvent trial design and leverage the use of naturalistic stimuli by adapting the forward linear model to compute an equivalent of ITPC (and PAC, shown later on) for continuous M/EEG recordings. We could indeed reproduce those results of word-related modulation by computing a TRF model with word onset only on band limited data (in Fig. 3a, b). Again a cluster-based permutation test revealed a significant difference against baseline (taken as the average signal for negative lags up to -50 ms). We further investigated the peak times of the evoked activity contributing to the phase clustering in the low-frequency cortical activity (inset topographies in Fig. 3a). This suggests a fronto-temporal wave propagating from posterior to anterior

sensors in the delta-band and in the opposite direction for the theta-band, such low-frequency travelling waves have been recently proposed as a mechanism for temporal binding and integration[63] as well as for encoding structural information[56].

These preliminary results indicate the presence of modulation in the canonical frequency bands, thus validating our choice of frequency bands for further analysis. Note that given the report of theta-gamma coupling in the literature for speech processing[46,48,55], we are adding the gamma band to subsequent analysis. We then proceeded to investigate the influence of linguistic features on the phase and power modulation of the MEG signal within the following bands: delta

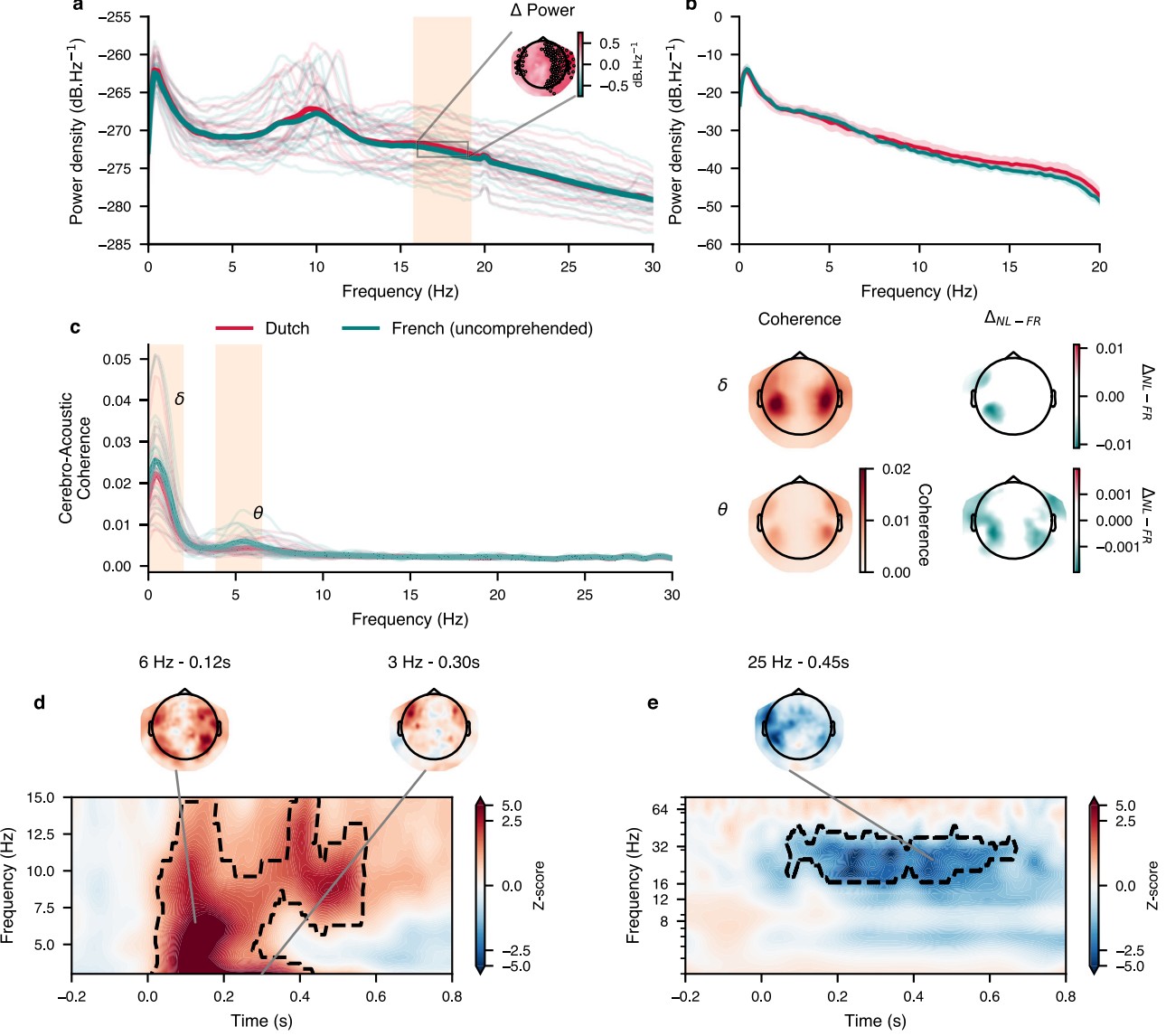

**Fig. 2 | Power spectral density (PSD) of the MEG data and the stimuli, and word-related phase and power modulation. a**, **b** PSD of MEG averaged over sensors for French and Dutch listening conditions. A cluster-based permutation test revealed no significant difference between average power, although the permutation revealed a marginally significant cluster in the beta band, showed with the shaded area (*p*-value = 0.06). The topographic inset presents the power difference within this frequency band, marked sensor are sensors for which the difference were significant (one sided paired *t*-test with α = 0.05, dof = 24, fdr-corrected for comparisons across 269 sensors). **b** PSD of the acoustic envelope average across stories within each condition, no significant difference found (using cluster-based permutation using independent *t*-tests as cluster statistics). **c** Cerebro-acoustic coherence. We computed the magnitude squared coherence between MEG sensor data and speech envelopes. The shaded areas are clusters with a cluster *p*-value

below 0.05 (*p* = 0.00977 and *p* = 0.00098 for delta and theta-band respectively). Note that the coherence is actually greater for the French condition. The first column of topographic plots on the right indicate average coherence values in the given regions; in the second column we show the difference contrast between Dutch and French conditions within those frequency bands. **d**, **e** Inter-trial (time-locked on word onsets) phase clustering (ITPC) and power modulation respectively, averaged across sensors. The contour outlines significant time-frequency cluster (cluster *p*-values of 0.001 and 0.005 respectively, cluster-based permutation using one sample two-sided *t*-test, cluster threshold at t(1−0.005, 24) = 2.8, applied on data after baseline removal, therefore testing for difference w.r.t baseline). Using a similar statistical approach, we did not find any significant difference between French and Dutch listening conditions.

(0.5–4 Hz), theta (4–8 Hz), beta (15–25 Hz), gamma (30–80 Hz). We computed the TRF models for each linguistic feature set and frequency band, and evaluated the reconstruction accuracy of the models. The results are presented in Fig. 3.

## Joint contributions of rule-based and statistical features
We now assess how brain signals vary in response to specific aspect of linguistic stimuli. Namely, we built two rule-based features, both derived from syntactic constituency trees: *close* designates the number of phrasal constituents a given word is closing (thus counting closing

brackets at each word), while *depth* stands as a proxy of ongoing syntactic complexity simply by measuring how deep in the hierarchy a given word is. The two other features, *surprisal* and *entropy*, are derived from a neural language model from which we can compute the probability distribution of next word prediction based on the preceding context. We first analysed the spatio-temporal dynamics of low-frequency activity in response to all features using a convolutional model which consists in estimating so-called temporal response functions (TRFs). We assessed the reliability of those features in representing the MEG data by computing the reconstruction accuracy

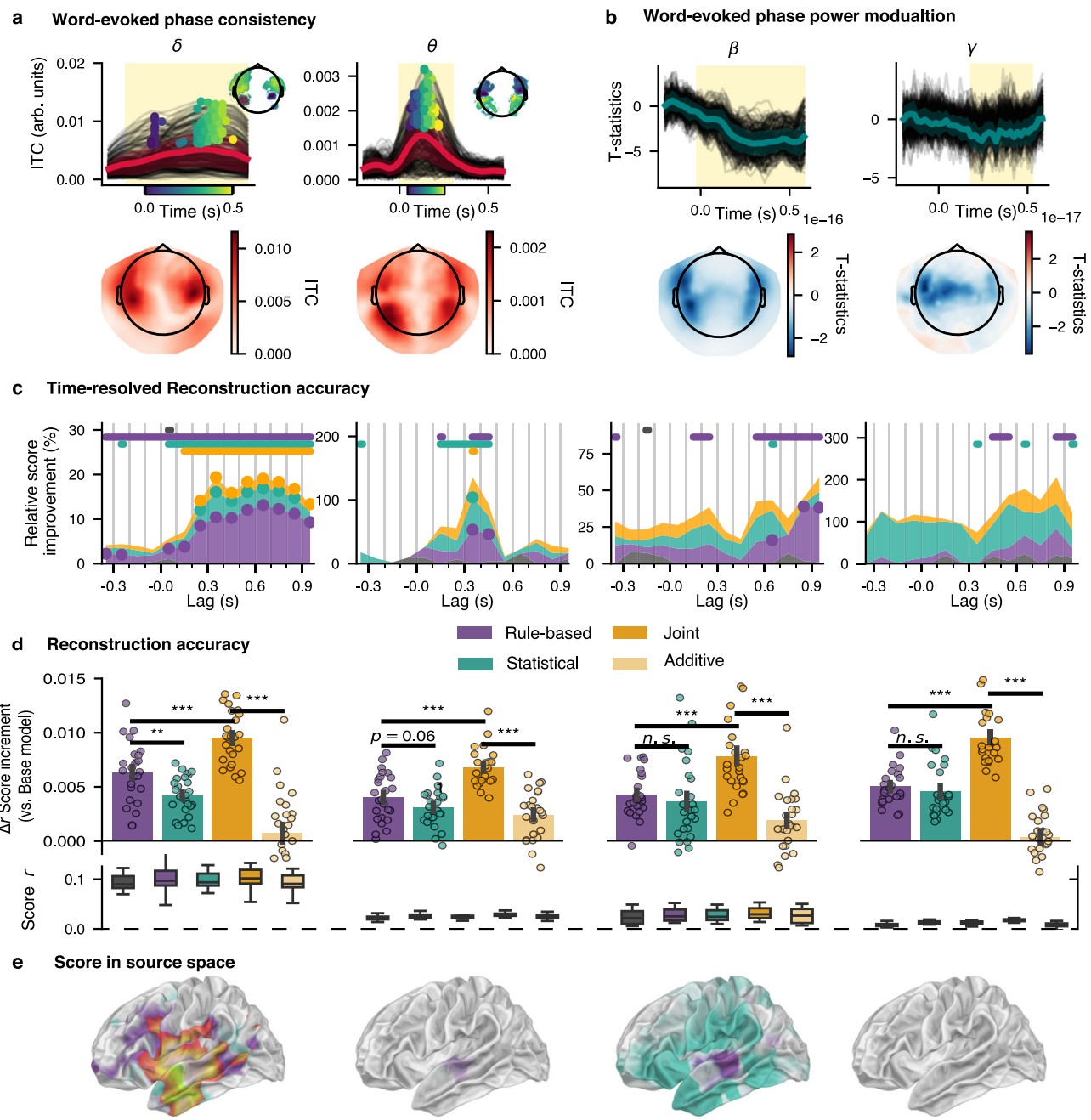

**Fig. 3 | Phase and power modulation. a** Inter-trial Phase Clustering (ITPC) in delta (left) and theta (right) bands computed using a TRF model with only word onsets on band passed normalised complex data (so a signal of the form $e^{i\phi(t)}$). In bold red is the time course of the cluster of sensors for which we found significant effect using cluster-based permutation compared against baseline (negative lags). The shading yellow area indicates the time span of the cluster. The peak of each sensor time course are marked according to their respective latency (colour scale for latencies shown below the x-axis), those peak times are also reported on the topographic inset. **b** Power modulation computed on band-limited power using a TRF model with word onsets only. **c** Reconstruction accuracy for different time windows, with relative contribution of each feature set, displayed as a stacked area plot. Thick segments on top show uncorrected significant time intervals (one sample two-sided t-test against a population mean of 0, n = 25), circle markers indicate significant time intervals corrected for multiple comparison (Bonferroni,

n = 13 time windows). **d** Mean scores increment (over base models scores) for each feature sets and frequency bands. Error bars show standard error of the means. Significance evaluated from paired t-tests (two-sided, dof = 24) comparing mean between feature sets. The bottom box plots present the absolute score distributions. Box boundaries represent the quartiles of the data while whiskers extend to the smallest and largest values within 1.5 times the interquartile from the lower and upper quartiles respectively. The overlaid points show individual data (n = 25). **e** Unique (green and purple for statistical and rule-based sets respectively) and shared (red to yellow colour scale) contribution to score, projected in source space (one-sided paired t-test comparing the mean score at each vertex location, dof = 23). A transparency threshold corresponding to a corrected p-value of 0.05 (FDR correction for 8196 vertices) has been applied (note that for delta, the threshold is lower: $p < 1e^{-4}$).

at each sensor (or sources) for several models (as described in Table 1). As a baseline, we used a model with only acoustic and word onset regressors. Every model is then matched in number of regressors by swapping the feature of interest with a null feature, which follows the

same statistics. Each null model for a given feature set consists of a TRF computed by using a shuffled version of the stimulus features (the shuffling is strict, as we keep word onset intact and shift values of linguistic features by several words). For instance, to compute the

**Table 1 | Model names and feature sets used in the analysis**

| Feature | Model Name | | | |
| --- | --- | --- | --- | --- |
| | Base | Statistical | Rule-Based | Joint |
| Envelope | ✓ | ✓ | ✓ | ✓ |
| Word onsets | ✓ | ✓ | ✓ | ✓ |
| Surprisal | ✗ | ✓ | ✗ | ✓ |
| Entropy | ✗ | ✓ | ✗ | ✓ |
| Depth | ✗ | ✗ | ✓ | ✓ |
| Close | ✗ | ✗ | ✓ | ✓ |

The ✓ symbol indicates that a given feature is present in the model while ✗ indicates that a null version of the feature was used (shifted values while keeping the same onset timings).

relative increase in reconstruction accuracy of Rule-based features (as seen in Fig. 3d), we alter the values of the *close* and *depth* features while keeping their original timings. By comparing the score to baseline null models, we normalise for the increasing number of features as each null model contains the exact same number of regressors. As seen in Fig. 3d, each set of features generates a significant increase in reconstruction accuracy compared to null models. A linear mixed-effects (LMM) model analysis, including subject variability as a random intercept with a random slope per frequency band, revealed that the feature set had a significant overall effect on the relative score (likelihood ratio test against LMM without feature set factor, $\chi^2(17) = 231.65$, $p < 0.001$) and that frequency band does not significantly influence score outcomes ($\chi^2(18) = 15.53$, $p = 0.63$), suggesting that the pattern seen for each model score might be similar across frequency band. Importantly the full model, which contains both statistical and syntactic features, shows a significantly higher reconstruction accuracy than both other models. We also compared the reconstruction accuracy of the joint model, comprising all features, and a model where we combined TRF coefficients from the two independently trained "statistical" and "rule-based" models (called additive model in Fig. 3). The rule-based model presents a marginally higher reconstruction score compared to the statistical-only model in the delta band (paired $t$-test on means, $p < 0.005$, correcting for multiple comparison with FDR (Benjamin/Hochberg)). No significant differences were observed in other frequency bands. Across every frequency bands, the joint model performed better than either specific models (all tests in Fig. 3d are controlled for multiple comparison with FDR (Benjamin/Hochberg) correction on $p$-values).

**Time-resolved contributions of syntax and statistical features**
We further investigated the time-resolved contribution of each feature set to the overall score. We computed the score for each feature set for a series of time windows, by evaluating a TRF model *per* window of analysis. Therefore, there is a distinct TRF model trained for small non-overlapping segments of 100ms each, spanning from -400–900 ms around word onsets. This is shown in Fig. 3c for each frequency band, where the stacked plots represent the relative increase (in percentage) of the model of interest over an acoustic model of reference. The base model uses only speech envelope as a feature, while other features are mismatched between stimulus and MEG. Results show that syntactic features mostly contribute to the score throughout a large time window extending at earlier and later time lags. For both feature sets, we see significant reconstruction beyond and above acoustic and word-onset only models (Fig. 3c shows uncorrected significant segments with a coloured bar above time regions (paired $t$-tests, dof = 25). We then adjusted $p$-values of our statistical tests using a conservatory Bonferroni correction, therefore dividing our critical $p$-value by the number of lags and feature set, to control for potential increase in type I errors. Corrected significant time segments are shown with a marker on the corresponding time-course).

For this time-resolved analysis, we used causal filters in order to avoid any spurious artefacts due to filtering which could alter the score of earlier lags (note that for the rest of the analysis we chose to keep anti-causal FIR linear filter in order to not alter the phase of signals). Despite this, we observe significant improvement at negative lags in the delta-band. In a supplementary analysis (Supplementary Fig. 2) we compared the score for each individual feature without grouping them. This showed that *entropy* and *close* contribute to the reconstruction accuracy at negative lags. This is not anti-causal as the entropy at a given word depends only on the previous words heard. In other words, it is possible to observe a neural response time-locked to the current word onset if we assume that the timing of the coming words is anticipated. For *close* feature, the same reasoning does not hold, however it can be argued that, in many cases, words closing constituent phrases are also likely to be more anticipated. This can also be an evidence of anticipatory mechanism for syntactic processing of ongoing structure building. The auditory system have been shown to be sensitive to long-term duration prior, learned from lifelong exposure over language-dependent syntactic cues[64]. Finally, the improvement in reconstruction accuracy for syntactic features at later time lags is likely due to the integration of words into larger syntactic units. This is supported by the fact that TRF time course for closing bracket count feature present the largest coefficients at later time-lags. This feature corresponds to the number of syntactic units that are closed at a given word, which is a proxy for the integration of words into larger syntactic units.

**Phase amplitude coupling**
It has been suggested that the phase of low-frequency activity may modulate power of higher-frequency band limited activity or oscillations in order to align excitability of neural population with relevant segment in speech[48,61] or in order to temporally bind and encode structural information[56]. The question remains whether this coupling between low- and high-frequency neural activity is also modulated by the predictability of words, thus affecting the temporal prediction of syllable and word onset[53]. We decided to investigate the presence of phase-amplitude coupling (PAC) in the MEG data, and in particular to disentangle the effect that distinct linguistic features may have on PAC.

First, we examined the presence of PAC across the entire signal, regardless of the timing of words. That is, we used equation (5) but instead of averaging across trials, we took the average across time. We computed PAC modulation index for each label in source space following the parcellation from Destrieux et al.[65] to limit computational time and memory (74 labels per hemisphere). We found evidence of PAC between the phase of delta-band activity and the power in the beta and gamma range, as well as between the phase in the theta range and gamma power (Fig. 4). PAC estimates were compared to surrogate data in order to compute a normalised metrics by z-scoring the resulting modulation index using the mean and standard deviation from the surrogate data computation. Those standardised metrics were then contrasted with the French condition for statistical testing. We found significant PAC when averaging over labels with the strongest mean PAC values (paired $t$-tests, dof = 24, realised per phase and amplitude frequency bins, fdr-corrected to control for multiple comparisons). We observed greater PAC over fronto-temporal network for both delta-beta and theta-gamma PAC (Fig. 4, bottom panels).

Furthermore, we are introducing a new method to disentangle the effect of different stimulus features to PAC. As such, we are investigating PAC relative to word onsets. The simulations described in Fig. 6 show that we can recover the effect of an individual feature affecting phase-amplitude coupling using a linear forward model based on the complex analytical signal formed with the power of high-frequency activity and the phase of low-frequency activity $r_{high}(t)e^{i\phi_{low}(t)}$. This novel approach relies on a linear forward model akin to temporal response function computation where instead of the band-passed

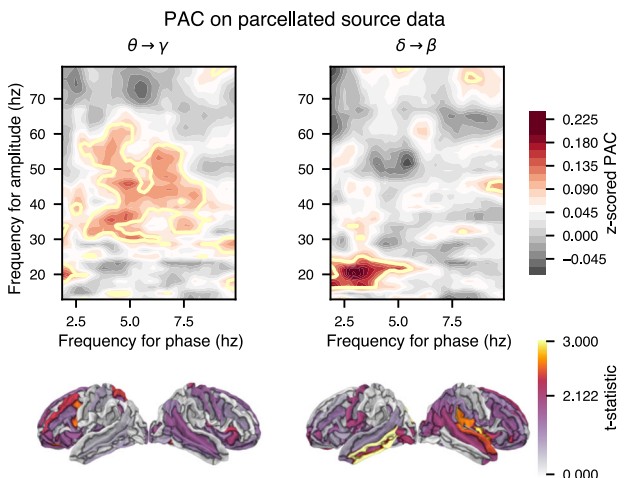

**Fig. 4 | The top panels present the stimulus-wide Phase-Amplitude Coupling (PAC), not time-locked to word events.** PAC values were normalised (z-scored) against PAC estimated from surrogate data (shuffling amplitude segments). As every frequency across phase and amplitude are scanned, PAC was only computed for labels of a parcellation of the source model to reduce computation time and memory resources. Yellow outlines indicate phase-amplitude bins for which FDR-corrected $p$-values, correcting for multiple comparison across frequencies and time bins, are < 0.05 (one-sided $t$-test, dof = 24, testing for population mean > 0.0). The bottom row shows the raw t-statistic (dof = 24) of PAC strength (Dutch - French) across participants at the source level. Note that the parcellation used[65] is outlined with grey lines.

MEG data we used the normalised analytical signal (see bottom panels of Fig. 1). By doing so, we effectively look at how the phase of the complex analytical signal clusters at a given lag. The advantage of this method is that it takes into account every predictor we feed into the forward model and thus allows us to establish which features weigh in the most (see details in Methods section, simulations of this technique are presented in Fig. 6, and further discussion can be found in the supplementary materials).

We computed the PAC for each feature set and compared it to a null model where only the feature of interest is being shuffled. First, the likelihood ratio test comparing LMMs with and without the categorical variables for feature sets and frequency bands indicated a significant overall effect ($\chi^2(7) = 378.63$, $p < 0.001$), suggesting that these factors contribute to variations in the relative score. Pairwise analysis revealed greater score reconstruction for the joint model for delta-beta while the Rule-based model presents stronger score in theta-gamma (see Fig. 5a and c, paired $t$-tests, dof = 25, $p$-values adjusted with FDR correction, to compare score increment w.r.t Base model between each model: for the delta-beta PAC: between Statistical and Rule-Based model, $p = 7.5e^{-4}$; between All and Rule-based model $p = 2.7e^{-9}$; for theta-gamma: $p = 0.014$ and $p = 0.48$). To investigate the effect of PAC from a specific feature, we computed null models where only the feature of interest is being shuffled, one at a time. Then we used a permutation cluster-based analysis to compare the coefficient of the PAC for a given feature and its corresponding null model. Fig. 5 (panels b and d) show the resulting phase-amplitude coupling coefficients for individual features (supplementary Fig. 3 shows all features, including controls, such as PAC from acoustic envelope). We summarize the PAC coefficient by computing the global field power (mean squared value) across sensors. A cluster-based permutation test was run against baseline mean values (t-statistics as test statistic, with primary threshold of $p < 0.05$, randomly permuting condition labels 1000 times, significance threshold for clusters set at 0.05). All features significantly modulated cross-frequency coupling in the delta to beta bands. While only precision entropy and closing bracket counts

showed a significant modulation of gamma power through theta phase. Higher PAC coefficients were found in the superior temporal gyri bilaterally. In the left hemisphere, we found clusters in the inferior frontal gyrus, the anterior temporal lobe (for *close* only) and the superior temporal gyrus. The left inferior frontal gyrus and the left anterior temporal lobe are two regions that have been previously associated with syntactic processing and semantic composition[27,58,66].

## Discussion

A growing number of studies demonstrated how to measure the sensitivity of the brain to naturalistic speech for word-level features. It remains difficult to control for confounding aspects, such as in Ding et al.[24], where it has been argued that the chunking observed around syntactic phrases might be elicited simply by word-level occurrence statistics or by the repetition of part-of-speech tags[25].

We built a word-level representation of naturalistic speech encompassing both syntactic, rule-based, features and statistical, data-driven, features. We showed that both feature sets could be recovered from MEG signals and that they could be used to decode the comprehension state of the subject. Finally, we showed how phase-amplitude modulation jointly occurs for both feature sets, suggesting that they are both processed and orchestrated in parallel. We found overlapping brain regions for both feature sets, in particular in the left inferior frontal gyrus, and in the anterior temporal lobe, two brain regions that have been previously associated with syntactic processing and semantic composition[26,27,66]. Using naturalistic stimuli, we dissociated, on the one hand feature set computed solely based on statistics of word sequences as computed via GPT model and, on the other hand a set of rule-based abstract features built directly from constituency tree structures. Temporal response functions obtained from those feature sets could each explain variance in MEG signals beyond chance and gave enhanced representations of the signal that allowed for decoding of comprehension state on unseen subject data. Nelson et al.[27] showed that broad-band gamma activity was more sensitive to syntactic surprisal, dissociating statistical features from syntax. Although, they trained statistical models purely on sequences of part-of-speech tags, which conveniently remove the lexico-semantic properties of the context. In the present study, we cannot control for the syntactic information contained in the statistical features, but we verified that the sensitivity to phrasal boundaries is at least minimal in those features. In particular, the language model is trained to predict the next word given the previous ones, and thus may learn an approximation of syntactic distributions across words as much as that approximation facilitates next word prediction. Therefore, the statistical features are not fully independent of syntax. However, syntactic features are computed from annotations of the constituency-parse tree structure of the sentences, and thus do not contain any information about the statistics of word sequences. And we assume that they are solely driven by the syntactic structure of the sentences.

The result of the joint model explaining over and beyond any of the individual feature sets suggests that the brain is sensitive to both aspects of language in an inseparable manner. This synergy is observed in regions where both types of information are encoded with equal strength. We found that both feature sets consistently drive an increase relative to our base model (envelope and word onset) across frequency bands (see Fig. 3). Only in the delta band we observed a significant improvement of Rule-based feature over Statistical ones. Syntactic processing has been previously shown to elicit a stronger phase-locked neural activity within the delta-band in response to speech[24,49,67,68]. However, as both feature sets contribute to a significant increase, we infer that statistical information aids the inference of categorical information. The idea that both prediction errors and representational states, which in the present study may be related to statistical and syntactic features respectively, are processed hierarchically and in parallel at different levels has been recently the

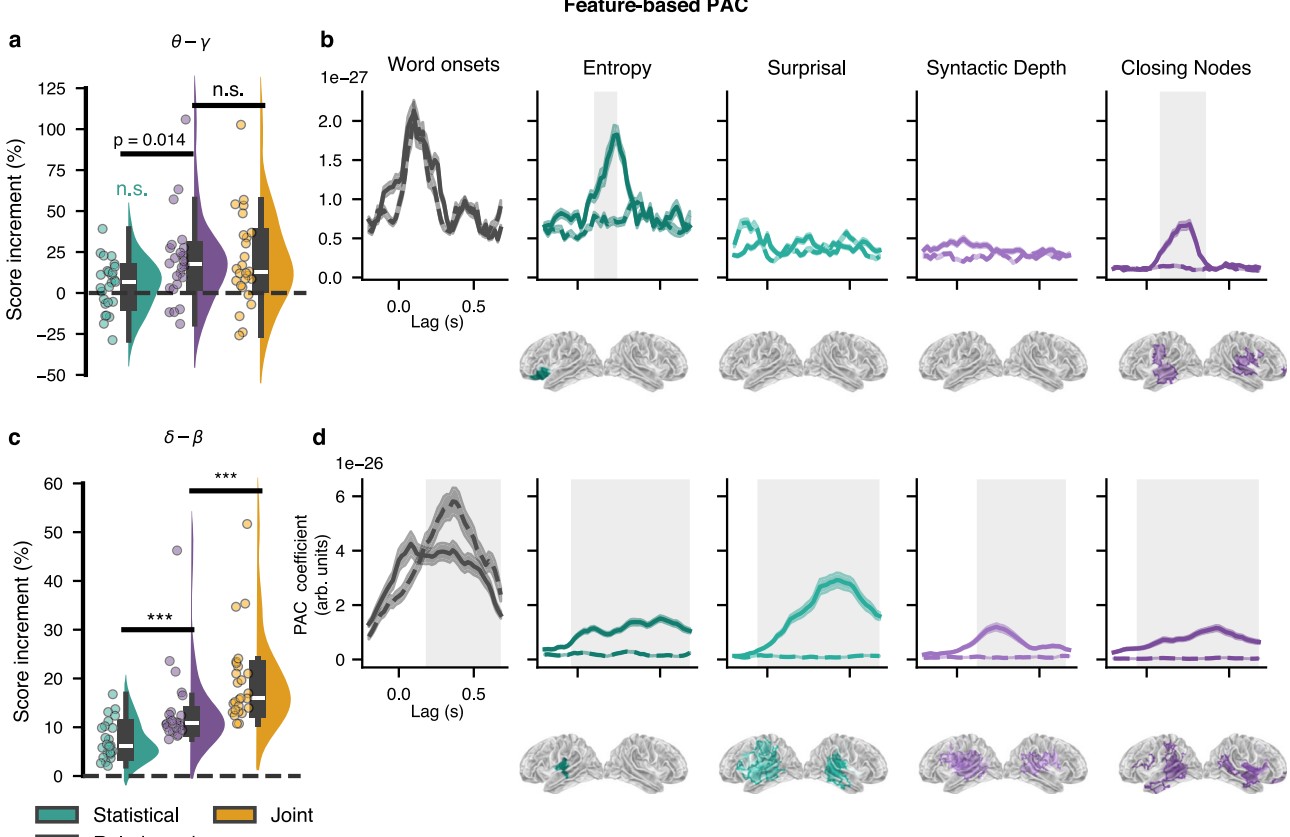

**Fig. 5 | Feature-based Phase-Amplitude Coupling (PAC). a, b** Theta-gamma phase amplitude coupling. **c, d** Delta-beta phase amplitude coupling. Left panels show the increment of reconstruction accuracy with those Temporal Reponse Function (TRF) models against the Base model (envelope and word onsets). Two-sided paired $t$-tests were used to compare the scores of the different models (dof = 24, stars indicate a quantisation of the exponent in power of 10 of the $p$-value). The box plots present the data distributions. Box boundaries represent the quartiles of the data while the whiskers extend to the smallest and largest values within 1.5 times the interquartile from the lower and upper quartiles respectively. In the right panels (time course and source space), cluster-based permutation test (using two-sided one sample $t$-test as a statistic, initial threshold set at the $t$-value for $\alpha = 0.05$ with dof = 24) was used to compare the PAC coefficient values of a given feature and its corresponding null model. The time course represents the global field power across sensors (standard error of the means shown as shaded area around curves). Grey shaded areas present temporal spread of significant clusters. After projecting in source space, we ran a spatio-temporal cluster based permutation to test significance of coefficients against baseline. We show, below the time courses, the time-averaged cluster in source space for each feature, masking the observation statistics below the 97.5th percentile of their distribution.

converging result across a number of studies using naturalistic stimuli for speech[11,14,69] This effect can be observed at other levels of granularity during speech processing, for instance, in the processing of the spectro-temporal acoustic features into categorical phonemes, and from phonemes themselves into lexical representations - in this sense, integration from phonemes to words, is modulated by statistical cues from top-down predictions[5,13,59]. In particular, our findings relate to Giraud & Arnal[5] who showed that slow cortical activity is related to the processing of the predictability of the next word in a sentence. Our results extend this finding by showing that low-frequency phase information modulates gamma power in a way that is related to the predictability of the next word in a sentence, namely when the uncertainty of those predictions is high.

Different aspects of syntax have been previously linked to brain activity ranging from syntactic violation and the P600 event-related potential component[70] (for a review, see Kuperberg, 2007[71]), to phrase structure tagging[24] and syntactic surprisal[27]. In particular, the *merge* operation[26,27,72] has the privilege of reflecting a crucial aspect of syntax, namely its recursive and combinatorial power. The merge operation is fundamental for composition and syntactic unification. It is also crucial in endowing language with its recursivity, allowing it to generate an unbounded set of utterances from a finite vocabulary set. Researchers have focused on pinpointing a specific location for such

computation[26]. However, it is unclear whether such computation, in the brain and in complementary fashion to computational-level claims in formal theories, occurs in isolation from semantic processing and more generally, from the dynamical update and integration of cues at different levels of the speech and language processing hierarchy[21]. We used the closing bracket count as a proxy for the binding operation, similar to other studies[23,27]. We found that spatio-temporal responses to both syntactic features are generally left-lateralised, but not exclusively so; moreover, we found distributed activity in the left anterior temporal lobe and in parietal regions and inferior frontal gyrus. In the time domain, we found a significant effect of closing nodes in both delta and theta bands. Other studies have attributed neural tracking to speech in the theta-band to the active chunking process of word units[73], perhaps aided by predictions[5,59]. In that respect, our data suggest that syntactic processing may also play a role in aligning with linguistic units and in providing a basis for tracking of such linguistic units.

Syntactic and predictive processing do not need to be fully disjoint in the brain. Each feature can jointly contribute to the prediction of upcoming input while at the same time help in integrating context with currently processed units. Locally, in a given neural circuit, we propose that binding of neural representations can be made through phase-amplitude coupling. This mechanism has been proposed to be a

general mechanism for neural binding[74,75], and has been argued to be involved in the processing of linguistic features[5,56,59,61,67]. In particular, we found that the coupling between theta phase and gamma amplitude was stronger, specifically for entropy and for the integration of nested trees. This may suggest that the two feature sets are processed in parallel. These features reflect how neural dynamics anticipate precision-weighted predictions and integrate a word to the current phrase respectively.

We see evidence of phase alignment and power modulation from the word-evoked activity by measuring phase constituency and induced power across all canonical frequency band of M/EEG. Importantly, the phase constituency may reflect mere evoked and time-locked activity, and not only oscillatory mechanisms. Therefore, we do not take those results as evidence for the presence of oscillatory response for low-frequency signals. Beta band power might, however, be reflecting oscillatory power change. Interestingly we see a typical decrease in beta power at the word-level. But once we disentangle the effect of specific features, we find for instance, a positive effect of syntactic depth, a broad proxy for syntactic complexity, which is in line with the structure and sentence-building effect leading to increased beta power observed[72,76,77]. More recently, an analysis on the dependency parse of sentences on this very same dataset also provided with evidence for the role of beta for maintenance and/or prediction during dependency resolution. Zioga et al.[30] showed that the beta power was modulated proportionally with the number of dependencies to be resolved. This would result in a consistent power increase while complex sentences are being processed. We also find a positive effect of depth in the delta band, which is in line with the syntactic processing account of delta tracking[16,24,49,54,67,68].

Our results support the dual timescale for predictive speech processing as proposed by Donhauser & Baillet[59]. The observed theta/gamma coupling occurs for highly expected information gain (high entropy leading to stronger coupling) and for words which must integrate into a larger number of constituents. This is in line with the idea that theta is a sensory sampling mechanism tuned to maximize expected information gain[59]. A word that closes several nested constituents will generally be well predicted syntactically (for its part-of-speech tag is constrained by context) but highly anticipated semantically (anticipated, as in expecting high-information content although not actually known or predicted), as it plays a crucial role in the sentence (and even more possibly in word-final languages). As such, an externally driven, weakly entrained, oscillator could synchronize the integration of a word into a larger syntactic structure while aligning through phase synchrony and nested theta-gamma oscillations the excitability of neural assemblies engaged (as in[48] though at word level here). This is synchronized both with the input and the internally generated temporal predictions further as the expected information gain is high. The computational model proposed by Ten Oever & Martin[53] also supports such framework where the predictability of a word will generate a corresponding phase advance or lag. Although in our study, it is precisely the expected information gain that is driving the phase synchrony. On the other hand, delta/beta coupling occurs for words that are less predictable, i.e., with a high surprisal value. This is in line with the idea that delta is encoding non-redundant information as in Donhauser & Baillet[59]. In other words, novel information deviating from internally generated predictions generates an update of the internal model. Top-down predictions, but also updates, have been already linked to beta power modulation[61,76]. All in all, this mechanism, if supported via delta-phase, is bound to endogenously generated predictions and thus to the internal model of the listener[21]. Given the synchronous increase in coupling for syntactic features (depth and close) we suggest that this internal model update is aligned with the slower rhythm of (predicted) phrasal boundaries[60]. We find these results to align with the idea of multiplexed and alternating predictive and integrative mechanisms between, respectively, top-

down and bottom-up information as seen in Fontolan et al.[78] or in the syllabic-level speech inference model from Hovsepyan et al.[61].

If based purely on data showing tracking to sound envelope, we must note that delta band does not reflect purely linguistic processes. First we see on the present data that coherence with sound envelope is stronger in delta for uncomprehended language, and that power alone does not carry sensible difference between our listening condition (Fig. 2a, b). Moreover, low-frequency tracking of envelope is sensitive to acoustic changes too, rather than linguistic information, for instance it becomes larger when pauses are inserted in the stimulus[79]. Here however, we note the role of delta phase and its coupling with higher frequency band, in particular beta, for the processing of linguistic features. We observed strong dependency of beta modulation, coupled through the phase of delta due to word surprisal. This is in line with the idea that beta carries out top-down predictions[56,61] and internal model updates while being modulated by the phase of endogenously generated delta rhythms[30,77,80]. Given the importance of theta- and delta-band speech tracking for comprehension[38,45,62,81], we propose that the coupling between delta and higher frequency bands is a general mechanism for the processing of linguistic features, possibly leveraging the temporal regularities of lower-level cues to align with internally generated predictions and representations. Previous studies have highlighted the role of beta oscillations for predictive coding[61,82].

We use surprisal as a proxy for information prediction, which may or may not take some syntactic and semantic features into account. It certainly does not build upon lower levels such as spectro-temporal acoustic information, as this would imply having trained a statistical language model from raw audio data. In the classical view of predictive processing though, bottom up prediction errors are matched against top-down predictions. One caveat in language is that, beyond the sensory representations, the hierarchy of representation is not well-defined in the brain. Structural and semantic information may not need to always sit within a strict hierarchy (except for example when function application or domain specification for semantic functions is specified by syntactic structure - which may be quite often), and the brain may nonetheless leverage both sources of information to work in parallel. Thus, it is unclear what becomes top-down or bottom-up information at the sentence level, when considering only word-level features. Moreover, we must account for the temporality of speech, where contextual information, from previous words, also helps to refine predictions. In this study, surprisal and precision (entropy) encompass such information from the word sequence, while syntactic features are more likely stemming from higher levels sources (i.e., not purely lexically-driven) if we consider the possibility that sentence level representations serve as top-down predictions for upcoming word-level processing. Indeed, the information added by incoming words within nested syntactic structures comes about by taking in account the embedding into a larger constituent, and thus reflects linguistic knowledge and processing over the entire sentence, if not discourse. As such, on a predictive account, the information from lower levels, such as phoneme sequence and fine-grained acoustic information, together with their relation to word recognition, must also be accounted for to fully capture the predictive context of speech processing. A comprehensive account of a predictive coding theory of speech processing must therefore incorporate an ongoing model of language processing together with perceptual processing of incoming sensory input.

In conclusion, we investigated the contribution of rule-based syntactic features together with the sequential predictability of words to cortical signals. Each set of features resulted together in a more accurate representation of MEG signals, suggesting an overlap in how the brain encodes and processes those different features. By computing a forward model, we could extract the neural response to those features from naturalistic listening conditions, thus leveraging the

need to manipulate the stimulus to exhibit responses to phrase structures. Syntactic operations, such as *merge*, operationalised here with the closing bracket count, together with the depth in constituency tree structures, showed a temporally broader response around word onset as compared to the sequence-derived features of surprisal and entropy. Across several frequency bands we observed distinct networks with some overlapping regions were both feature sets improved reconstruction accuracies. In particular, we found that the statistical features were more sensitive to theta band activity, while the syntactic features were more sensitive to delta band activity.

Phase-amplitude coupling analysis revealed that the two feature sets are processed in parallel, and are used to predict the next segment (lexicalised items in the current study) as it is integrated with the current context. We found that the coupling between theta phase and gamma amplitude was stronger, specifically for entropy and for the integration of nested trees. This suggests that the two feature sets are jointly used to update predictions and integrate the linguistic content to the current phrase. Given the role of theta band in speech tracking[59,62,73] we suggest that such rhythms are tightly bound to the acoustic signal and reflect thus temporal predictions aligned with high expected information[53]. More generally, all features presented significant cross-frequency coupling between delta and higher frequency bands. But we observed a larger delta-beta coupling for surprisal, potentially in line with the idea that beta carries out top-down predictions and internal model updates while being modulated by the phase of endogenously generated delta rhythms. Given the importance of theta- and delta-band speech tracking for comprehension[38,45,62,81], we propose that the coupling between delta and higher frequency bands forms a core cortical computational mechanism for the processing of linguistic features, as speech becomes language, and leverages the temporal regularities of lower-level cues to align with internally generated predictions and representations.

## Methods

The study was approved by the ethical commission for human research in Arnhem and Nijmegen (CMO2014/288). A total of 25 participants (18 women, between 18 and 58 years old) completed the experiment. Informed consent was obtained from all individual participants. All participants were right-handed native Dutch speakers with no reported fluency in French despite incidental exposure. Participants self-reported their (in)ability to understand a sentence in French. Participants were given monetary reimbursement for their participation.

### Experimental design

Participants were asked to listen and pay attention to several audio stories while we simultaneously recorded their MEG. The stimulus consisted of Dutch short stories (from Hans Christian Andersen and the Brothers Grimm) and French stories (from Grimm, E. A. Poe and Hans C. Andersen) available online on the Gutenberg project, totalling 49 min and 21 min, respectively. All stories were divided into short story parts lasting between 5 and 7 min (leaving a total of 9 Dutch story parts and 4 French ones). Each story part was presented without interruption, while participants fixed a cross in the centre of the presentation screen. Participants were prompted on-screen with five multiple-choice questions between each part to assess their attention and comprehension. Stimuli were presented using the Psychtoolbox library on Matlab[83].

MEG data were acquired at 1200 Hz using a CTF 275-channel whole-head system (VSM MedTech, Coquitlam, Canada) in a magnetically shielded room. The MEG system was equipped with 275 first-order axial gradiometers with a baseline of 5 cm. The head position was measured before and after the experiment using five head position indicators (HPI) coils. The HPI coils were activated every 200 ms

during the experiment to monitor head movements. The head position was corrected for each story part using custom-built Matlab code to display how the head aligns with the initial recorded position; participants were asked to adjust their head position to fit head markers if the movement was too large. Participants could self-pace the start of a trial after answering the behavioural comprehension question, allowing them to pause between stories blocks. Each participant also had a structural MRI scan (T1-weighted) using a 3T MAGNETOM Skyra scanner (Siemens Healthcare, Erlangen, Germany). The MRI scan was used to reconstruct the cortical surface of each participant using FreeSurfer software (Martinos Center for Biomedical Imaging, Charlestown, MA, USA). The cortical surface was used to project the MEG sensor data onto the cortical surface using a linearly constrained minimum variance beamformer[84]. Finally, we measured the head shape of each participant using a Polhemus Isotrak system (Polhemus Inc., Colchester, VT, USA) to co-register the MEG and MRI data.

### MEG preprocessing

The original MEG data were recorded at 1200 Hz. We first resampled the data at 200 Hz after applying and anti-aliasing low-pass filter to the data. Noisy channels and flat channels were marked as bad for interpolation and to be discarded in subsequent analysis (computation of covariance or ICA algorithm). We removed blink artefacts by matching ICA component time courses to measured EOG and similarly removed heartbeat artefacts. Note that the ICA decomposition was run on data filtered between 1 and 40 Hz, we then kept the ICA spatial filters for subsequent processing while discarding the band-passed data used to compute them. Finally, we applied a notch filter at 50 Hz to remove line noise.

**Source reconstruction.** We then used all data to compute a data covariance matrix, and used it to compute a noise-normalized linearly constrained minimum variance (LCMV) beamformer[84]. The LCMV beamformer was created using a 7 mm grid with 3 mm spacing, using functions from the MNE-python library[85].

**Time-frequency analysis.** was performed on the MEG data using the MNE software package[85]. We first epoched the data around word onset, with no baseline applied. We then used the Morlet wavelet transform to compute the time-frequency decomposition of the MEG data. The Morlet wavelet transform was computed with varying number of cycles from 2 to 7 cycles. We used a logarithmically spaced grid of frequencies between 3 and 80 Hz, with 32 steps. The time-frequency decomposition was computed for each epoch. We extracted the average power by taking the absolute of the complex Fourier coefficients and the inter-words phase constituency by summing the complex Fourier coefficients after normalising them by their absolute value.

### Stimulus representation

Most of the analyses performed relied on linear models. We computed forward encoding models which map stimulus features to MEG data. Such models are also called temporal response functions (TRF)[86]. This approach gives us a way to assess the importance of each feature in the stimulus in explaining the MEG data. The obtained model coefficients are also directly interpretable in terms of modulated neural activity, as opposed to filters learned from backward models[87]. In this section, we will first present how each stimulus features were defined and computed, and then how the TRFs were estimated from the data.

In light of the current literature, we focused on rule-based syntactic features on the one hand, following cortical tracking of hierarchies suggested by Ding et al.[24], and on statistical features reflecting predictive processing of sequences[38]. The syntactic features were computed using the Stanford parser[88]. The statistical features were computed using the GPT2 language model[89].

Importantly, we do not assume that language processing is supported by information-theoretic metrics in the hard sense. That is, we are not proposing a theory of language understanding from mere surface statistics. However, following the hypothesis given by theories on predictive processing in various cognitive domains and notably for perception, we assume that the brain can extract information-theoretic features from the stimulus, regardless of the specifics of its representational format (e.g., functionally semantic or syntactic), and that these features are actively used to predict internal representations. Those information-theoretic features are thus a proxy for the underlying domain-general cognitive process of predictive inference, notably for predictive processing and Bayesian inference, where surprisal becomes a proxy for prediction error[59,90] and entropy for uncertainty[3]. Both of these are critical concepts in predictive processing. These quantities are involved in most predictive mechanisms[4,90], where cues are extracted at different representational levels to predict or infer upcoming linguistic information.

On the other hand, rule-based features are derived from specific instantiations of parsing mechanisms (such as constituency trees in context-free grammar), which already support a particular theory of syntax derived from linguistics rather than neuroscience. Again, we are not suggesting by the use of those particular features that the brain is precisely implementing such a parsing strategy to compute syntactic representations. Nevertheless, the known sensitivity to such structures[24] motivates the use of a metric that captures the general shape of syntactic trees throughout naturalistic sentences. We decided to create a set of features that tracks the complexity of such structures and also captures the integrative mechanisms at play when words or phrases need to be integrated into a larger syntactic unit. We note that although the left-corner parsing strategy has been previously suggested as a better model for cognitive processing of syntax[22,58,91], this was done on English data and potentially differs in head-final or mixed word order languages[16,92]. Finally, it may be that for the method applied here, the fine-tuning of parsing strategy does not matter as much as the mere presence of brackets, indicating constituent structures[16].

- Rule-based (or Syntactic) features: To build our syntactic features, we first ran a tree parser on every sentence from our stimuli.
    1. Depth: Syntactic depth is a proxy for syntactic complexity. A word highly embedded within nested structures, which is to say, a word deep in the tree structure hierarchy, will carry a higher value for this feature. This may reflect several cognitive processes among which the maintenance in working memory of syntactic structure or the general cognitive load for processing complex sentences.
    2. Close: This refers to the number of subtrees being closed at a given word. Some words do not close any subtree, and some will close several at once. This feature encompasses the variability accounting for integrative mechanisms such as "merge"[26]. When words or phrases need to be grouped into a larger syntactic unit this feature is incremented. It is also referred to as *bottom-up* count of syntactic structures[16,23,92]. This is in contrast to the top-down count, which enumerates opening nodes. Giglio et al.[92] found bottom-up parsing to better represent structure-building during comprehension rather than production, which we decided to use here.
- Statistical features: Extracting the information-theoretic values requires an estimate of the probability distribution of each word in the utterance conditioned on the previous word (over the entire vocabulary). This was quantified using GPT2, a state-of-the-art language model, which is trained specifically to causally predict word from sequences of textual data.
    1. Surprisal: The negative natural logarithm of the probability of a given word item conditioned on the sequence of preceding words; $-\log(P(w_i|w_{i-1}, w_{i-2}, \ldots, w_1))$.

2. Entropy: It quantifies the amount of uncertainty, at a given word, in predicting the next word. Mathematically, this is the expected surprisal, which we can also interpret as the expected information gain. It is computed as $-\sum_{\text{words}} P(w_i|w_{i-1}, \cdots) \log(P(w_i|w_{i-1}, \cdots))$

On top of those features of interest, we used two other regressors to control for the low-level acoustic and prosodic effects on neural signal as well as any unexplained variance at the word-level. This was achieved by using the acoustic envelope along with a "word onset" features. The former was computed from the audio waveform by half-rectifying it and then applying a low-pass filter (type I FIR filter, $f_c = 20$ Hz, transition bandwidth of 10 Hz, attenuation in stop band of 60dB, ripples relative to peak $= 1e^{-3}$) and raising the signal to the power of $\frac{1}{3}$ to mimic the non-linear compression from early auditory processing stages. The latter comprised of a comb-like feature with constant values of "ones" aligned with every word onsets. A table of the features used in the analysis is presented in Table 1.

The correlation is not excessively high and should not, presumably, lead to collinearity issues in the subsequent analysis. To verify we computed the Variance Inflation Factor (VIF) for each feature. All VIFs were below 2, which is below the threshold of 5, indicating that there is no collinearity issue in our data[93].

## Temporal response functions

With the speech representation time-aligned to the magneto-encephalographic recordings, we then compute optimal filters that map the stimulus to the MEG signals. This is known as forward modelling, or encoding, and in particular the present method boils down to extracting *temporal response functions* of the above features. This method assumes a convolutional linear model mapping from stimulus to MEG data. Such an approach has been successfully applied to recovering brain responses to sound envelope[7,62], allowing to further decode auditory attention, and also in estimating brain responses to ongoing linguistic features[38,94]. This has been applied with different neuroimaging data such as fMRI, ECoG, and M/EEG.

We model the MEG response signal at sensor $i$ (or source estimate location) $\{y_i\}_t$ as a convolution between a kernel $\beta$ (to be estimated) and the stimulus representation signal $\{x_j\}_t$ for the $j$th feature. Activity from the data that is not captured by this model is supposed to be Gaussian noise.

$$y_i(t_n) = \sum_{j=1}^{N_{feat}} (\beta_{ij} * x_j)(t_n) = \sum_{j=1}^{N_{feat}} \sum_{\tau=1}^{\tau_{\max}} \beta_{ij}(\tau) x_j(t_n - \tau) + \epsilon_{in} \quad (1)$$

$$\hat{y}_i(t_n) = \sum_{j=1}^{N_{feat}} \sum_{\tau=1}^{\tau_{\max}} \hat{\beta}_{ij}(\tau) x_j(t_n - \tau) \quad (2)$$

Note that the hat symbol, ˆ, correspond to the estimated or reconstructed values. It is easy to rewrite equation (2) in a vectorized form as $\hat{\mathbf{Y}} = \hat{\boldsymbol{\beta}} \mathbf{X}$ where the temporal dimension is expressed as column vectors and each channel data (vector) are concatenated along the row dimension of the matrix $\hat{\mathbf{Y}}$. In this formulation, $\mathbf{X}$, called the design matrix in the context of linear models, contains all the lagged time series of every feature in its columns and time samples along its rows: $\mathbf{X} \in \mathbb{R}^{(N_{feat} \cdot \tau_{max}) \times N_{samples}}$. The matrix $\hat{\mathbf{Y}}$ contains the reconstructed MEG data, with the same number of rows as $\mathbf{X}$ and as many columns as MEG sensors or source estimates: $\hat{\mathbf{Y}} \in \mathbb{R}^{N_{samples} \times N_{sensors}}$. Finally, $\hat{\boldsymbol{\beta}}$ is the matrix of estimated TRFs, with a row for each lag for each feature and as many columns as MEG sensors or sources: $\hat{\boldsymbol{\beta}} \in \mathbb{R}^{(N_{feat} \cdot \tau_{max}) \times N_{sensors}}$.

Written in its vectorized form, we can easily see how the closed-form formula to estimate $\hat{\beta}$ arises:

$$\mathbf{Y} = \mathbf{X}\boldsymbol{\beta}$$
$$\mathbf{X^T Y} = \mathbf{X^T X}\boldsymbol{\beta} \qquad (3)$$
$$\hat{\boldsymbol{\beta}} = (\mathbf{X^T X})^{-1}\mathbf{X^T Y}$$

However, when the inversion of $\mathbf{X^T X}$ is unstable, which easily happens with continuous predictors, a regularised autocorrelation matrix is used instead (via Thikonov regularisation):

$$\mathbf{w} = (\mathbf{X}^T\mathbf{X} + \lambda\mathbf{I})^{-1}\mathbf{X}^T\mathbf{y} \qquad (4)$$

where $\lambda$ is the regularisation parameter, and $\mathbf{I}$ is the identity matrix. Wherever used, the regularisation parameter was set to $\lambda = <\lambda_k>_k$, the arithmetic mean over the eigenvalues of $\mathbf{X^T X}$.

All the TRF estimation and scoring was carried out using custom-built Python code. This allowed to optimise the computation of the TRF: in particular, we aggregate the computation of correlation matrices across stories and leverage on the use of singular value decomposition to compute the pseudo-inverse of the design matrix.

**Model evaluation.** We used Pearson's correlation coefficient to assess the quality of the model fit. We computed the correlation between the predicted and the observed MEG data for each model and each sensor or source location. We then averaged the correlation across sensors to obtain a single value for each feature set. In order to avoid overfitting, we used a leave-one-story-out cross-validation scheme. We trained the model on all but one story and then tested the model on the left-out story. We repeated this procedure for each story and then averaged the correlation across stories.

Moreover, we used mismatched feature values to compute a null distribution of scores and coefficients (see Table 1 for a description of each model used). We then compared the observed correlation to the null distribution to assess the significance of the model fit of a given feature set. Those mismatched features were generated by keeping the same onset timings but shuffling the feature values across words. By doing so, we are keeping the same temporal structure and statistics within the feature set but destroying any potential relationship between the feature and the MEG data. As an example, when we compute the score of a given feature set, e.g. Statistical features, we compare the observed score to the distribution of scores obtained by fitting the model with mismatched statistical features, while keeping other features intact. This method also has the advantage to control for the number of features in the model thus correcting the bias in the score, in particular, due to extra features when those are not explaining variance at a given sensor.

**Inter-word phase coherence and phase-amplitude coupling**
**Global PAC.** A first analysis for estimating phase amplitude coupling consisted in scanning across several frequencies using a time-frequency representation of the MEG data. We used the Morlet wavelet transform to compute the time-frequency decomposition of the MEG data. The Morlet wavelet transform was computed using seven cycles between 1 and 80 Hz. We extracted the average power by taking the absolute of the complex Fourier coefficients and the phase by normalising the complex wavelet output by its absolute value. We used the python package `Tensorpac` (version `0.6.5`)[95] to extract this global standardised PAC coefficient. The library's PAC computation also dealt with the shuffling of amplitude segments to compute the surrogate data.

**TRF-based PAC.** One challenging aspect in computing the inter-trial phase consistency (ITPC, or inter-event phase clustering and in the case of the present study: inter-word phase clustering) is that it relies on the clustering of phase, evidently, across trials (defined by word onset timings). Hence, it depends on trial-based experimental design. We propose an innovative approach to circumvent trial design and leverage the use of naturalistic stimuli by adapting the forward linear model to compute an equivalent of ITPC and PAC for continuous M/EEG recordings.

Our approach consists in using the complex analytical signal to compute a complex-valued TRF which will jointly contain phase and amplitude information. Computing the TRF is equivalent to finding the kernel of the convolution, which triggers the recorded response and thus captures and summarises the phase concentration from a continuous recording. In essence, the TRF computed from a comb-like time-series as features is analogous to ERP analysis. We can extend this analogy and consider this method as a way to compute ITPC and PAC from continuous recordings.

The classic, trial-based, ITPC is computed as:

$$ITPC_i(t) = \frac{1}{N_{trials}}\left|\sum_{trials} e^{\phi_{i,\delta}(t)}\right|$$

While the PAC can be computed as[74]:

$$PAC_{\delta \to \beta}(t) = \frac{1}{N_{trials}}\left|\sum_{trials} r_\beta(t)e^{\phi_{i,\delta}(t)}\right| \qquad (5)$$

Therefore, we can use the continuous signals $e^{\phi_{i,\delta}(t)}$ and $r_\beta(t)e^{\phi_{i,\delta}(t)}$ to compute ITPC and PAC respectively, with respect to different features. Using only word onset (a feature that is one at word onset and zero otherwise) is equivalent to the trial-based computation of those quantities. However, this extended framework allows us to incorporate other exogenous variables and thus analyse their respective contributions to ITPC and PAC. Forte et al.[96] have successfully applied such a modelling approach on complex signals to recover phase and amplitude response in brainstem recordings.

In order to investigate whether this method reliably recovers features that specifically contribute to some phase-amplitude coupling, we simulated the effect and recovered the dynamics of coupling from features, beyond the event-related mean vector length. Those simulations are presented in Fig. 6. We simulated three different scenarios: (A) a simple phase-amplitude coupling where the phase of low-frequency signal modulates the amplitude of a faster signal regardless to event timing. Then we simulated phase-amplitude coupling occurring in a time-locked manner after some events. Finally, a third simulation was performed where phase-amplitude coupling occurs in a time-locked manner and the amplitude is modulated differently depending on the values of an external feature signal. We then computed the TRF for each of those simulations and compared them to the mean vector length computed across trials. As seen in Fig. 6, we can recover the effect of an individual feature affecting phase-amplitude coupling from the linear forward model.

**Statistics**
Power spectral densities, coherence, ITPC and power modulation comparisons (in Fig. 2) were carried out using cluster-based permutation tests[97] implemented in MNE Python[85]. We used 1024 permutations and a threshold of $p < 0.05$ to determine significant clusters.

Reconstruction scores and kernel coefficients learned through linear regression (TRFs) were tested for significance against a null model of temporal response functions which contains control regressor nonetheless (Base model). We constructed an empirical estimate of the null distribution represented by a model where syntactic and predictive features were unrelated to the stimulus. This is realised by shifting the values of each feature while keeping the word-onset timings intact. Therefore, each reconstruction score can be

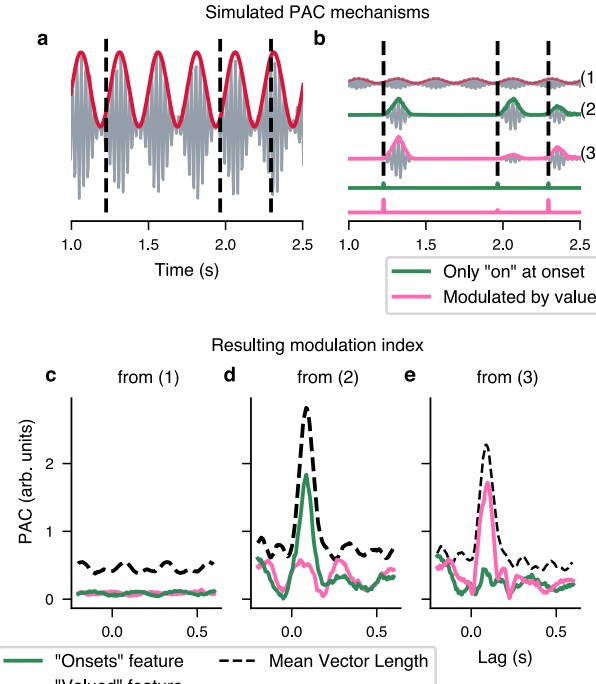

**Simulated PAC mechanisms**

**Resulting modulation index**

"Onsets" feature — — Mean Vector Length
"Valued" feature

**Fig. 6 | Simulation of feature-dependent Phase Amplitude Coupling (PAC) and estimation thereof. a** Simulated phase-amplitude coupling: The left panel shows a simple phase-amplitude coupling where the slow phase of the signal in red modulates the amplitude of a faster signal in green. **b** We simulated three scenarios of PAC mechanism: (1) reproduces the global PAC effect of (**a**) without any time-locking to events, in the other two examples the amplitude is modulated differently depending on the timing (2) and values (3) of an external feature signal. **c**–**e** We computed the modulation index for PAC using either equation (5) (mean vector length, black dashed line) or the TRF-based method described in the main text (green and pink lines). The computation is done on each of three PAC simulations from (**b**). **d**, **e** show that we can recover the effect of an individual feature affecting phase-amplitude coupling from the linear forward model. On the other hand, the mean vector length method is unable to disentangle if the PAC is induced by onsets only (green line) or valued features (pink).

compared to the corresponding null model which matches the number of features. To assess the overall effect of model structure (Rule-based, Statistical or Joint) and frequency band, we ran a linear mixed model (LMM) analysis with the relative score difference with respect to the Base model as a dependent variable and a random intercept per subject (and random slope per frequency band where applicable). This was done using `statsmodels` library (version 0.14.1) with the formula: `score ~ feats + fband + (1 + fband | subject)`. We then determine the significance of fixed-effects with a likelihood ratio test with a nested model which only account for intercepts. Finally, post-hoc analysis on the relative score differences is done using one sampled *t*-tests, comparing their value against zero. The resulting *p*-values were corrected for multiple comparisons using the Benjamini-Hochberg, controlling for the false discovery rate (FDR), implemented by MNE Python's `fdr_correction` method[85].

Global PAC in Fig. 4 are computed by standardising the computed modulation index with one obtained from surrogate data (by shuffling amplitude segments). We then compared thos standardised coefficient between Dutch and French condition using paired *t*-tests, correcting for multiple comparisons with FDR correction after averaging across labels with the strongest PAC within each frequency band.

For TRF time-courses (PAC coefficients, in Fig. 5b, d bottom row), we used a cluster-based permutation tests to assess significance

(against baseline mean value) and extract spatio-temporal clusters[97] using MNE python's *permutation_cluster_1samp_test* with 1024 permutations and using the default threshold value except when indicated in the main text. In the case of the comparison of reconstruction scores in the source space (Fig. 3e), we used a paired *t*-test to compare the scores across feature sets in a pairwise manner. The resulting *p*-values were corrected for multiple comparisons with FDR correction and used to display the significance of the comparison in source space.

### Reporting summary
Further information on research design is available in the Nature Portfolio Reporting Summary linked to this article.

### Data availability
The raw MEG data generated in this study have been deposited in the Radboud University Repository database and are openly available with the identifier https://doi.org/10.34973/a65x-p009[98]. Processed MEG data and Source data underlying the figures in this paper are available in Figshare with the identifier https://doi.org/10.6084/m9.figshare.24236512.

### Code availability
The code supporting the findings of this study is available on the GitHub repository at https://github.com/Hugo-W/feature-PAC[99]. All the analysis was carried out using Python 3.12 using custom code, heavily based on usage of the MNE-Python library (version 1.6.1)[85]. Figures are done using Matplotlib (version 3.8.3) and Seaborn (version 0.13.2).

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

## Acknowledgements

A.E.M. was supported by an Independent Max Planck Research Group and a Lise Meitner Research Group "Language and Computation in Neural Systems", by NWO Vidi grant 016.Vidi.188.029 to AEM, and by Big Question 5 (to Prof. dr. Roshan Cools & Dr. Andrea E. Martin) of the Language in Interaction Consortium funded by NWO Gravitation Grant 024.001.006 to Prof. dr. Peter Hagoort. Hugo Weissbart was supported by NWO Vidi grant 016.Vidi.188.029 to AEM. We thank the members of the Language and Computation in Neural Systems Group for assistance and support in the form of team data collection and discussion of our results.

## Author contributions

H.W. and A.E.M. conceptualized the study. H.W. designed and implemented the experiment, collected and preprocessed the data; developed the code and analysis of the data. H.W. and A.E.M. wrote, reviewed and edited the manuscript.

## Funding

## Competing interests

The authors declare no competing interests.
