## [Peer Review File · Nature Communications]

The Structure and Statistics of Language Jointly Shape Cross-frequency Neural Dynamics During Spoken Language ComprehensionReviewer #1 (Remarks to the Author):

This study examines the representation of structural and statistical features of language in neural dynamics, measured with magnetoencephalography (MEG). The authors use state-of-the-art methods to reconstruct oscillatory dynamics from these features. They show that both structural and statistical features contribute to reconstruction accuracy, which is taken – together with spatial and temporal overlap of the corresponding dynamics – as evidence that these features are not processed completely separately in the brain.

I found this an ambitious study using sound and sophisticated methods to advance the field. I also appreciated the detailed data analysis, yielding several interesting results. However, I believe that the description of hypotheses and results can be improved.

I found the introduction relatively long, covering several, relatively complex topics. Perhaps as a consequence, it was not fully clear to me on which ground previous work had claimed “a strict separation of linguistic structure and statistics in the brain”, and how precisely results from the current work can disprove the claim. Is it enough to show that both types of features contribute to reconstruction accuracy? Is spatial overlap sufficient to claim that there is no strict (functional) segregation? I believe that this could be explained better, in combination with a description of the background that is more compact and focused on the hypothesis.

Similarly, it was not always evident how different results described can address this hypothesis. To be clear, I’m not saying that they cannot; it could just be pointed out more directly. Emphasizing this issue, I found it sometimes hard to associate findings described in the text with individual panels in the figures. Panel labels and those of the y-axis are missing in figures 1 + 2 and 3A,B respectively, and the references to the figures in the text could be more precise.

Together, a more compact and structured description of hypotheses and results would address my concern.

Other points:

- I found the topographies difficult to interpret as many sensors are outside of the head, is there an alternative visualisation?

- An overview of stimulus characteristics might be helpful, in particular a spectrum of the speech envelopes. Can differences between languages explain some of the results shown in Figure 1? I might be mistaken but the two languages seem only contrasted in Figure 2(A-C), and I wonder whether this contrast is also important for results shown later. Shouldn’t participants’ ability to extract linguistic features affect results?

Reviewer #2 (Remarks to the Author):

The paper by Weissbart and Martin presents a novel approach for exploring how cortical oscillatory patterns generated by naturalistic speech stimuli can be related to linguistic

information present within the speech signal – namely syntactic structures and contextual semantic dependencies. In doing so they demonstrate a complex set of results, highlighting the contribution of semantic and syntactic linguistic features to both cortical phase clustering and phase-amplitude coupling across multiple frequency bands.

I believe that the novel method of extending ITC and PAC approaches to naturalistic non-trial-based datasets using extension of the mTRF methods is a worthwhile contribution to the literature which opens new methodological avenues. Reporting of the modelling procedures (fig.7) is particularly appreciated. Nonetheless the paper in its present form (given the complex nature of both analysis and the findings) does not make completely clear for the reader the theoretical contributions of this work. I outline my related methodological and theoretical concerns below.

Major points:

1. The introduction of the paper sets out to assess the contribution of semantic-distributional (as referred to 'statistical') and the syntactic features in speech to the cortical oscillatory dynamics induced by the speech signal. While the choice of the semantic features is clearly supported by the literature (semantic surprisal and entropy) in the introduction the choice of the syntactic features (number of brackets, depth of the syntactic tree) seems more arbitrary and perhaps overly simple given other metrics that have been shown to have cognitive validity/ greater explanatory power for language-related behavioural responses: e.g. metrics generated by left-corner parsers (c.f. Oh, Clark, and Schule, 2022).
2. I would strongly suggest including the speech envelope into the models tested with the TRF approach. As authors discuss in the introduction, it is often difficult to distinguish the contribution of syntactic versus acoustic characteristics of speech to the cortical oscillatory dynamics generated by the speech stimulus. Especially given that landmarks of syntactic structures are often correlated to salient acoustic events marked by intonation fluctuations. Hence it is critical to show that models that include syntactic features have explanatory power over and above the models that include only word onsets and envelope.
3. There are two methodological concerns I have related to the use of the TRF approach. First, I wanted to clarify that when comparing models to the 'null models' only one feature at a time has been permuted/'shuffled' in time. This means that, for example, significance of syntactic feature contribution has been asserted by comparing the full model (word onsets + semantic + syntactic features) to the null model where only the syntactic feature has been permuted (word onsets + semantic features + permuted syntax). The model fit of the full model over and above (change in r) the same model with only one feature permuted isolates the contribution of a specific feature while controlling for presence of other contributing features (see Broderick et al., 2021). From the methods text and paper figures it is not clear if just one feature at a time has been shuffled (for the null models) while others have remained constant.
4. My second methodological concern is model fit at the negative / before 0 ms lags (see fig. 3 panels C and D) for the different frequency bands that authors relate to predictive processing of the linguistic features. A known feature of the mTRF approach is regression artefacts at the edges of the selected temporal window for the analysis – see original toolbox paper Crosse et al., 2016 for the explanation. Hence the window taken for the analysis is typically 50 ms larger to exclude those edge artefacts. Is it possible that these early increases in model fit are driven by the edge artefacts?
5. It is not entirely clear why individual features were grouped into sets when evaluating ITC and PAC feature contributions and between-model comparisons (e.g., the Rule-based vs

Statistical features model effect comparison). Semantic Entropy and Surprisal are thought to be linked to distinct cognitive processes (semantic predictions versus prediction errors) and similar arguments can be made for syntactic features (number of brackets versus depth in the tree). It is likely that only one of the two grouped features contributes significantly to the strength of the response (at a given lag) for a given oscillatory pattern. How do results look when those features are not grouped?

6. The inconsistency in model comparisons further impacts the clarity of the interpretations in the discussion. Specifically, the claim (in the discussion) that different features (semantic/syntactic) have distinct effects on the oscillatory patterns/frequency bands. This statement is critical for the discussion but seems not so be fully supported either by the ITPC or PAC results for individual models. For ITPC (fig. 3) it is not clear if the reconstruction scores or the model-fit (r) can distinguish between different semantic and syntactic features. For PAC the results seem clearer (Fig.4,5,6) but it is not clear that statistical/semantic features are preferentially modulating theta-gamma coupling and syntactic by delta-beta. For instance, robust semantic/statistical Surprisal effects are driving delta-beta coupling, and the effects seems more extended across lags and source space than either of the Depth or Close models. Where the spatiotemporal courses for individual feature reconstitutions (Fig. 5,6.) compared directly (e.g., spatiotemporal cluster permutation)? It would also be important to show that linguistic-features based model remain significant after modelling out the effects of acoustic envelope (see previous point). In the last section of the discussion authors acknowledge possible contribution of the envelope to low-frequency effects and argue for linguistic feature-related PAC effects and this interpretation will be strengthened by including the envelope into the models.

Minor:

1. There is something odd with Figure 2 – for me the power density (top plot, panels A B) and spectral modulation (plot G) appear empty hence it is not possible to evaluate those features of the MEG signal. I am not sure if something was wrong when uploading/formatting of that figure.

2. Figures overall need some cleaning up: please make sure the panel names (A,B,C) appear consistently in each figure and are similarly formatted. Some figures are missing axis labels (please double check) and label fonts are too small for some of the plots. The terminology for model names is not consistent (Rule-based/Syntax; Stats/Statistical).

Reviewer #3 (Remarks to the Author):

Title: “THE STRUCTURE AND STATISTICS OF LANGUAGE JOINTLY SHAPE CROSS-FREQUENCY DYNAMICS DURING SPOKEN LANGUAGE COMPREHENSION”

This study explores an interesting and important question in a rapidly raising domain of research. The authors used TRF and PAC analyses to study the neural encoding of word surprisal and sentence structure. The idea is generally interesting. It is indeed revisiting concepts that have been explored quite a bit in the literature, but I am not aware of other studies explicitly looking into these specific elements directly and simultaneously. So, I am of the opinion that the study is novel. However, I found the key results not convincing in the current version of the manuscript. Part of the problem is the manuscript had several quality issues, from the narrative that was missing crucial elements, to the figure, which are not of a sufficient

quality. This overall lack of attention to graphical details and explanations made the paper hard to read, potentially hiding some elements of interest. Please see my comments below. My opinion is that the implications of the proposed analysis will be useful in the corresponding domain of research, leading to many citations. Nonetheless, I consider this result fairly technical, and possibly not of interest to the broader readership of NCOMMS.

Major comments

1. The first major issue is methodological. I did not understand how the authors can claim that the effects they find are not due to the acoustics of the stimuli. Is that something I missed? Otherwise, the authors should run further analyses to verify that this is not the case (e.g., I suggested one way to do so in my comment about the TRF analysis below)
2. The second issue is the overall quality of the writing and figures. The narrative is jumping between sections with little explanations. The experiment is only explained in the Online Methods. Certain terms are defined too late and not used (e.g., TRF). The figures lack detail (as well as graphical attention e.g., font-size, missing labels). Statistical analyses are often only vaguely described.

Other comments

1. Line numbers are missing
2. Introduction: “However, language comprehension [...]”. That sentence is unclear. What does “not fundamentally predictable” mean exactly, and how is that consistent with the text in the parenthesis?
3. Last paragraph on that same page: It appears that only one research question is formulated, as the second point appears to be an operationalisation of the first question (i.e., how one would actually test the first question).
4. Introduction, page 2, paragraph 1: It is unclear to me whether this hypothesis is novel (maybe a few references in that part would help). And it is unclear how this hypothesis is derived exactly. Is this inspired from the large literature on syntactic and semantic violations?
5. Paragraph 2: This part is vague and it makes it sound as if the results mentioned in the references were all talking about “cortical oscillations”, which is not the case. A bit more structure would help this paragraph separating evidence on the relevance of certain EEG/MEG frequencies vs. the hypotheses on what “low-frequency [cortical] oscillations” reflect. (I note that this issue is appropriately mentioned later on in the introduction).
6. Results: First paragraph. I suggest replacing this sentence with a brief description of the analysis pipeline. I did not like that the paper jumps into the results without explaining the experiment.
7. Second paragraph. I think the claim that PSD is studied to assess the “presence of the expected neural oscillations” goes counter to the beautiful introductory paragraph on oscillations vs. evoked responses. I suggest adding a sentence to the introduction clarifying what the term “neural oscillations” refers to exactly. In my view (at this point of the reading), the authors are talking about the MEG power within a specific frequency-bin, rather than “neural” oscillations of an endogenous nature. A simple clarification of the specific use of “neural oscillations” would suffice. Or possibly changing that term to band-limited MEG power, or something similar.
8. A few comments on Figure 2: I don’t see the letters indicating the panels.. also, the mid-left panel is not showing the information appropriately, as the red lines are almost entirely hidden (maybe use transparency?). The y-label is missing. Mid-right panel: please rotate delta and theta. Also, the legends should include a name and legend. Are 4 different scales really

necessary? Font-size seems different across different panels. Mid panels: it's not entirely clear to me what those horizontal solid lines actually indicate. I got the point in the end (top row delta and bottom row theta), but do we really need such confusing lines. Instead, I suggest indicating delta and theta on the mid-left plot. Does the figure only show the actual values, or does any of those results include the result of the statistical significance test (a part from the top-left panel, which however is not explained appropriately in the caption – what is the yellow area exactly? $p < 0.05$? with what test?)

9. Overall, the second paragraph of the results is somewhat vague. More details clarifying how those results were actually obtained should be included. I appreciate the attempt of going straight to the point, but the authors should provide all the essential details to the reader at this point (especially on what exactly is tested in the statistical analysis). So, in sum, this paragraph should be clearer (e.g., the result for the mid-panels is unclear – where is the result of the cluster statistics shown exactly? Is it in the right panel? So, are those values thresholded by that statistical analysis?)

10. Figure 3: The authors should add an explanation of “depth close” earlier in the manuscript. Cosmetic comments: the font-size is absolutely inconsistent across the figure. Y-labels are missing. The meaning of the shading in A and B is not entirely clear to me. Panel D: datapoints appears to be hidden.

11. Figure 5: fix overlap of text on top-left. The authors seem to use yet another style for indicating the panels (lowercase, on the bottom of the panel). The figure caption is less than minimal. All the necessary details should be added instead (e.g., what do the shaded areas indicate exactly?). Again x- and y-labels are missing!

12. Section 2.1: Statistical results should be clarified and reported. Also, please refer to the specific panels when mentioning a figure.

13. I have a more fundamental question about section 2.1. Did the authors check that the increase is due to the actual information in the features rather than the larger number of parameters (e.g., by using shuffled features, rather than subsets of features)?

14. Section 2.1: The text should clarify that a temporal response function is estimated here with a lagged regression, or maybe deconvolution could be mentioned. There are many ways to explain this, but the current text (i.e., “linear forward model”) is insufficient. Also, the methods section should clarify which exact implementation is used (is it Eelbrain, the mTRF-Toolbox, or maybe something custom?)

15. From Figure 1, I gather that no acoustic feature was included in the model e.g., envelope, spectrogram. Why is that the case?

16. A related question: Could the increase mentioned in section 2.1 be due to acoustic differences captured by the “syntactic features”? This should be verified either by studying the stimulus itself and/or by considering the previous comment.

17. Section 2.3. The first sentence is empty. The only way to know what that means is to look at those references. Could the authors be more direct?

18. Methods. It seems that unnecessary filters were applied (for example, a low-pass and then a band-pass i.e., in that case, a low-pass and a high-pass would have been sufficient), which introduces additional unwanted artifacts that can be problematic (de Cheveigne and Nelken, Neuron, 2019).

Point-by-point response

The text from the reviewers is rewritten verbatim, using **bold** font weight and **sans-serif font**. Our responses are written in *italics and with serif*. References to the manuscript are written in **colourised text**, with a line number and section indicated in parenthesis.

Reviewer #1

This study examines the representation of structural and statistical features of language in neural dynamics, measured with magnetoencephalography (MEG). The authors use state-of-the-art methods to reconstruct oscillatory dynamics from these features. They show that both structural and statistical features contribute to reconstruction accuracy, which is taken – together with spatial and temporal overlap of the corresponding dynamics – as evidence that these features are not processed completely separately in the brain.

I found this an ambitious study using sound and sophisticated methods to advance the field. I also appreciated the detailed data analysis, yielding several interesting results. However, I believe that the description of hypotheses and results can be improved.

We appreciate your acknowledgement of our methods for exploring the neural dynamics of language processing through magnetoencephalography (MEG) and are also thankful for your recognition of the ambition behind our study and the approach we adopted.

We take your constructive feedback regarding the need for clearer hypotheses and results descriptions very seriously. In response, we have carefully revised our manuscript to articulate our hypotheses more explicitly and present our results in a manner that more effectively communicates the significance and implications of our findings.

I found the introduction relatively long, covering several, relatively complex topics. Perhaps as a consequence, it was not fully clear to me on which ground previous work had claimed “a strict separation of linguistic structure and statistics in the brain”, and how precisely results from the current work can disprove the claim. Is it enough to show that both types of features contribute to reconstruction accuracy? Is spatial overlap sufficient to claim that there is no strict (functional) segregation? I believe that this could be explained better, in combination with a description of the background that is more compact and focused on the hypothesis.

*Thank you for your insightful comments and suggestions. We acknowledge the need for a clearer presentation of our hypothesis and results. To address this issue, we have rewritten and reorganised the introduction and hope it is now better focused on the hypothesis. Notably, we have added the following sentences (l.51-58, Introduction): “**In light of this apparent dichotomy, and in the context of the debate in cognitive science regarding the role of statistical information in language processing [42-46], we then ask to what extent do their individual contributions explain neuroimaging data, is the whole better than the sum of its parts?. We hypothesise that they jointly contribute to explaining variance in the MEG data while presenting overlapping spatio-temporal sources. Moreover, the dynamics might disentangle them further as predictions and statistical inference seem to be a widespread phenomenon in cortical computation, while the organisation of linguistic units into nested hierarchical structures, at least at first blush, may be related to hierarchical processing in other domains in some ways, but not others [47-52]. We thus further hypothesise that brain responses to***

statistical and structural features are operated synchronously, with potential distinct time scales, and orchestrated through cross-frequency coupling.”.

We have also added a paragraph to the discussion to clarify the implications of our results for the hypothesis that linguistic structure and statistics are not segregated in the brain. However, we also highlight the limitations of that position more clearly. We indeed cannot draw such conclusions from simple spatial overlap and shared impact in improving reconstruction accuracy. Our main motivation and argument stem from the added value of using them in conjunction rather than in separate models. Indeed, we observed that having both feature sets in one model is better than either model alone, even when normalising for the increased number of regressors. We added another control to test for the gain of having a joint model with both representations. Indeed, we observe that compared to a model where we simply concatenate features fitted from independent models (so statistical and rule-based, separately), the joint model still performs better, suggesting a synergy between features in explaining neural data (Figure 3, panel d). So rather than arguing for no functional segregation, we argue, in line with the account of two parallel processes, for a synchronous processing regardless of whether it is spatially or functionally segregated. Moreover, we propose that a cross-frequency coupling mechanism may be pivotal in bringing both representations together, were they separated at a given point in time.

Our new introduction reads as follows (l.85-97, Introduction par.6):

Our hypothesis draws on the intersection of syntactic processing and predictive coding theories, positing that the brain's response to language is not just reactive but anticipatory, integrating both structural and statistical cues in real-time. We thus propose to link properties of syntactic structures, jointly with information-theoretic metrics to MEG data. Brennan and Pylkkänen [80] and Nelson *et al.* [26] have used a similar approach to link syntactic features to electrophysiological data. However, the former studies focused on the localisation of MEG activity to study word-evoked responses, while the latter analysed high gamma activity recorded from intracranial electrodes. In another study, Brennan & Hale [81] used information theoretic metrics built from context-free grammar parsers and delexicalised n-grams, which do not capture semantic information (thus their surprisal greatly differ from ours). Finally, in [22], a link is made between hierarchical syntactic features (node count) and surprisal from Markov models (n-grams, lexicalised and unlexicalised). While they elegantly show how different parsing strategies affect the prediction of neural activity, the statistics of word predictions are overlooked. Modern autoregressive language models produce a more precise estimate of conditional probabilities, from which we extract surprisal and also entropy, thus allowing for a better account of predictive mechanisms. To our knowledge, the current literature has not explored the interplay of such features across frequency bands with MEG.

Similarly, it was not always evident how different results described can address this hypothesis. To be clear, I'm not saying that they cannot; it could just be pointed out more directly. Emphasizing this issue, I found it sometimes hard to associate findings described in the text with individual panels in the figures. Panel labels and those of the y-axis are missing in figures 1 + 2 and 3A,B respectively, and the references to the figures in the text could be more precise.

Thank you for your comments, which have made our manuscript clearer and easier to follow. We have added panel labels to all figures and made sure that the references to the figures in the text are more precise. We now refer to figure panel labels directly and more precisely.

Together, a more compact and structured description of hypotheses and results would address my concern.

We rewrote the Introduction to be more focused and to set up direct links between our findings and the hypotheses. Your feedback has been invaluable in guiding these improvements. We hope that the revised manuscript will address your concerns.

Other points:

- **I found the topographies difficult to interpret as many sensors are outside of the head, is there an alternative visualisation?**

We understand the Reviewer's concern. This is a common visualisation for MEG data, while more typically, topographic maps of EEG data are interpolated only inside the head outlines. This is because the MEG headset is rather large, with around 300 sensors, so the ones located on the side of the head are meant to "wrap around" the head. This convention is adopted by the MNE-python toolbox (see their documentation of topomaps:

https://mne.tools/stable/auto_examples/visualization/evoked_topomap.html#additional-plot-topomap-options), which is the library we used for those particular plots. Nonetheless, we adapted those plots to slightly reduce the outskirts drawn, which we hope will reduce any possible confusion.

- **An overview of stimulus characteristics might be helpful, in particular a spectrum of the speech envelopes. Can differences between languages explain some of the results shown in Figure 1? I might be mistaken but the two languages seem only contrasted in Figure 2(A-C), and I wonder whether this contrast is also important for results shown later. Shouldn't participants' ability to extract linguistic features affect results?**

This was indeed intentional, as we eventually used other controls to simplify the interpretation of the results. For completeness, we have now added a comparison with the French condition to relevant result figures. We also added the power spectra of the stimuli in Figure 2 (panel b).

Reviewer #2

The paper by Weissbart and Martin presents a novel approach for exploring how cortical oscillatory patterns generated by naturalistic speech stimuli can be related to linguistic information present within the speech signal – namely syntactic structures and contextual semantic dependencies. In doing so they demonstrate a complex set of results, highlighting the contribution of semantic and syntactic linguistic features to both cortical phase clustering and phase-amplitude coupling across multiple frequency bands.

I believe that the novel method of extending ITCP and PAC approaches to naturalistic non-trial-based datasets using extension of the mTFR methods is a worthwhile contribution to the literature which opens new methodological avenues. Reporting of the modelling procedures (fig.7) is particularly appreciated. Nonetheless the paper in its present form (given the complex nature of both analysis and the findings) does not make completely clear for the reader the theoretical contributions of this work. I outline my related methodological and theoretical concerns below.

We sincerely appreciate the thoughtful and constructive feedback provided by Reviewer 2. Your recognition of the novel approach is highly encouraging. We are particularly grateful for your acknowledgement of our efforts to extend ITCP and PAC approaches to naturalistic, non-trial-based datasets. We acknowledge your concerns regarding the clarity of the paper's theoretical contributions and the complexity of the analysis and findings. In response, we have undertaken a thorough revision to articulate the theoretical framework better and clarify our methodological approach. We believe

these revisions will make our contributions to the literature more transparent and accessible to readers.

Major points:

- 1. The introduction of the paper sets out to assess the contribution of semantic-distributional (as referred to 'statistical') and the syntactic features in speech to the cortical oscillatory dynamics induced by the speech signal. While the choice of the semantic features is clearly supported by the literature (semantic surprisal and entropy) in the introduction the choice of the syntactic features (number of brackets, depth of the syntactic tree) seems more arbitrary and perhaps overly simple given other metrics that have been shown to have cognitive validity/ greater explanatory power for language-related behavioural responses: e.g. metrics generated by left-corner parsers (c.f. Oh, Clark, and Schule, 2022).**

We appreciate the reviewer's comment and agree that the choice of syntactic features could be better motivated. We now discuss these alternatives and their cognitive validity in the revised manuscript. We have added a sentence to the introduction to motivate the choice of syntactic features, adding relevant references (then further details follow in Methods and Discussion, see below) (l.30-33, Introduction):

Similar syntactic metrics have been used to study the effect of syntactic operations such as unification (the "merge" operation in the minimalist program [1,25]) or integration of an item into a larger structure or the depth, a proxy for ongoing complexity, of the syntactic tree at a given word [26-28]. These metrics align with foundational linguistic theories and have been shown to elicit predictable neural responses [22, 29-31].

Considering the constituency tree structure, these syntactic features are chosen for their direct interpretability and ease of computation. We believe that they are sufficient to demonstrate the potential of our approach. Moreover, they are directly comparable to the Merge operation found in the literature and used in other references. We also point out in our introduction (second paragraph, lines 20-23) that our study is comparable with the fMRI study by Brennan et al. 2016 who used surprisal (form n-grams) and node count (top-down and bottom-up though) and find it interesting that we can compare our MEG study to those results. This is mentioned in the Introduction (lines 16-22):

Traditionally, language's inherent unboundedness and generative aspect have often been seen as being in putative opposition to distributional and statistical accounts of language processing (e.g., [16, 17] for traditional view on the *competence hypothesis*; and [18,19] for grammar-free account of comprehension). In contrast to this dichotomy, in the present study, we synthesise these positions and present a framework wherein the syntactic structure and statistical cues are jointly processed during comprehension (see [20-22]). We build upon the work of Brennan et al. [23] who demonstrated the sensitivity of the BOLD signal to both structure and surprisal but focused on the localisation of such effect.

Moreover, other recent studies, in particular with Dutch language with MEG, showed that left-corner parsing node counts did not present a higher reconstruction score than top-down or bottom-up (this might in fact not translate to English, see discussion in the preprint from Coopmans et al, 2024). In another study from Coopmans in 2022, they found a response of neural signal to bracket count from a bottom-up parsing with surprisingly little difference to the mere presence of bracket (see their Fig. 6). We added text in the method section when describing our "depth" and "close" feature referencing those studies in contrast with Oh. J. et

al., 2022. Hopefully, this motivates further our particular choice of syntactic features. The updated methods section now reads (l.494-510, Methods):

We note that although the left-corner parsing strategy has been previously suggested as a better model for cognitive processing of syntax [28,81,132], this was done on English data and potentially differs in head-final or mixed word order languages as suggested by Coopmas *et al.* [52] and Giglio *et al.* [133]. Finally, it may be that for the method applied here, the fine-tuning of parsing strategy does not matter as much as the mere presence of brackets, indicating constituent structures, as seen in Coopmas *et al.* [52].

- Rule-based (or Syntactic) features: To build our syntactic features, we first ran a tree parser on every sentence from our stimuli.
 1. Depth: Syntactic depth is a proxy for syntactic complexity. A word highly embedded within nested structures, which is to say, a word deep in the tree structure hierarchy, will carry a higher value for this feature. This may reflect several cognitive processes among which the maintenance in working memory of syntactic structure or the general cognitive load for processing complex sentences.
 2. Close: This refers to the number of subtrees being closed at a given word. Some words do not close any subtree, and some will close several at once. This feature encompasses the variability accounting for integrative mechanisms such as "merge" [26]. When words or phrases need to be grouped into a larger syntactic unit, this feature is incremented. It is also referred to as *bottom-up* count of syntactic structures [23,52,133]. This is in contrast to the top-down count, which enumerates opening nodes. Giglio *et al.* [133] found bottom-up parsing to better represent structure-building during comprehension rather than production, which we decided to use here.
- 2. **I would strongly suggest including the speech envelope into the models tested with the TRF approach. As authors discuss in the introduction, it is often difficult to distinguish the contribution of syntactic versus acoustic characteristics of speech to the cortical oscillatory dynamics generated by the speech stimulus. Especially given that landmarks of syntactic structures are often correlated to salient acoustic events marked by intonation fluctuations. Hence it is critical to show that models that include syntactic features have explanatory power over and above the models that include only word onsets and envelope.**

We agree with the Reviewer that the inclusion of the speech envelope is critical for disentangling the contribution of linguistic versus acoustic features. Originally, we only used word onset as a control feature, with the idea of keeping only similar-type word-level features. We have now included the speech envelope in the models tested with the TRF approach. Importantly, adding this control did not change the pattern of results that we observed, strengthening the evidence for our conclusions.

3. **There are two methodological concerns I have related to the use of the TRF approach. First, I wanted to clarify that when comparing models to the 'null models' only one feature at a time has been permuted/'shuffled' in time. This means that, for example, significance of syntactic feature contribution has been asserted by comparing the full model (word onsets + semantic + syntactic features) to the null model where only the syntactic feature has been permuted (word onsets + semantic features + permuted syntax). The model fit of the full model over and above (change in r) the same model with only one feature**

permuted isolates the contribution of a specific feature while controlling for presence of other contributing features (see Broderick et al., 2021). Form the methods text and paper figures it is not clear if just one feature at a time has been shuffled (for the null models) while others have remained constant.

Your methodological concerns have highlighted areas for clarification in our TRF approach. The former analysis compared a full null model (all features but word onsets permuted) against the full model, while the model comparison of feature set were done without using permuted features. We have now harmonized the pipeline by following the Reviewer's suggestion. We now use the same number of features, permuting ("nullifying") only the feature(s) of interest. By doing so, we now effectively control for the number of regressors while properly isolating the contribution of a specific set of feature at a time. We explain this methodology in the new paragraph "Model evaluation" of the Methods section (line 552-560, Methods):

Moreover, we used mismatched feature values to compute a null distribution of scores and coefficients (see table 1 for a description of each model used). We then compared the observed correlation to the null distribution to assess the significance of the model fit of a given feature set. Those mismatched features were generated by keeping the same onset timings but shuffling the feature values across words. By doing so we are keeping the same temporal structure and statistics within the feature set but destroying any potential relationship between the feature and the MEG data. As an example, when we compute the score of a given feature set, e.g. Statistical features, we compare the observed score to the distribution of scores obtained by fitting the model with mismatched statistical features, while keeping other features intact. This method also has the advantage to control for the number of features in the model thus correcting the bias in the score, in particular, due to extra features when those are not explaining variance at a given sensor.

This approach is also explained throughout the Results section and in relevant captions. Moreover, we added a table in the Methods section (Table 1) to represent all our models.

- 4. My second methodological concern is model fit at the negative / before 0 ms lags (see fig. 3 panels C and D) for the different frequency bands that authors relate to predictive processing of the linguistic features. A known features of the mTRF approach is regression artefacts at the edges of the selected temporal window for the analysis – see original toolbox paper Crosse et al., 2016 for the explanation. Hence the window taken for the analysis is typically 50 ms larger to exclude those edge artefacts. Is it possible that these early increases in model fit are driven by the edge artefacts?**

This is a valid and important concern. The edge artefacts reported by Crosse et al. (2016) are not present in our (former) data analysis, as they only arise when using smooth continuous feature regressors. This is due to the inherent autocorrelation for neighbouring lags, which introduce degeneracy in the covariance of the time-lagged feature matrix ($X^T X$, which you can observe in the figure below as a "thickening" of the diagonal of this matrix) and forces the use of regularization. Edge artefacts then occur because of the background noise present in the response signal. Those artefacts can thus be interpreted as a form of overfitting for the discontinuity induced by noise at the edge of the kernel. In our case, we use discrete feature regressors ("comb-like" or "impulses"), which do not suffer from this issue. A short simulation

is included below to illustrate this effect.

Now, concerning the negative lags, we do not make specific claims about them, and we do not interpret the results of these lags. We present those results to highlight the relative importance of each feature at different lags in a broad sense. However, we now address this concern in the revised manuscript. There are two remaining reasons why an increased score can be observed at negative lags:

1. the specific feature indeed represents a predictive signal at negative lags, which is possible for some of our features, such as *entropy* and *close*. The former is a measure of the predictability of the upcoming word and, in essence, is not causally determined by the current word (as at a given word, its value only depends on previous words). The latter is potentially predicted by the representation of the ongoing tree structure. In either case, it should not be surprising to have an effect before 0ms for *entropy* (though potentially raising a point for discussion for “*close*”). We added a short explanation of this in the manuscript (l. 209):

This showed that *entropy* and *close* contribute to the reconstruction accuracy at negative lags. This is not anti-causal as the entropy at a given word depends only on the previous words heard. In other words, it is possible to observe a neural response time-locked to the current word onset if we assume that the timing of the coming words is anticipated. For *close* feature, the same reasoning does not hold, however it can be argued that, in many cases, words closing constituent phrases are also likely to be more anticipated. This can also be evidence of an anticipatory mechanism for syntactic processing of ongoing structure building. The auditory system have been shown to be sensitive to long-term duration prior, learned from lifelong exposure over language-dependent syntactic cues

2. there is ringing artefact from non-causal filtering inflating the score at negative lags. In order to address this concern, we reproduced all this timed TRF analysis with causal filters, and the results are consistent. The new analysis results are now included in the manuscript (Figure 3. Panel c).

However, to address another reviewer's comment and a comment made below, we added the speech envelope to all the TRF models, which will indeed contain edge artefacts. However, it is the only feature suffering from edge artefacts, and its effect is controlled as we only look at the improvement beyond the base model in each frequency band. Notably, the addition of the envelope did not change the pattern of results for those time-resolved score partitioning.

5. **It is not entirely clear why individual features were grouped into sets when evaluating ITPC and PAC feature contributions and between-model comparisons (e.g., the Rule-based vs Statistical features model effect comparison). Semantic Entropy and Surprisal are thought to be linked to distinct cognitive processes (semantic predictions versus prediction errors) and similar arguments can be made for syntactic features (number of brackets versus depth in the tree). It is likely that only one of the two grouped features contributes significantly to the strength of the response (at a given lag) for a given oscillatory pattern. How do results look when those features are not grouped?**

*We appreciate the Reviewer's comment and agree that the grouping of features need to be better justified. The main idea stems from the origin of those features. That is we group them following the perspective of the type model they are coming from rather than the (potential) cognitive process they elicit. This is mostly because the only true value we know about them is how we compute them rather than any hypothesis on what neural substrate may respond to each of them. That is, depth and close are derived from constituency trees, which themselves are built from a context-free grammar (hence "Rule-based"), while entropy and surprisal come from a statistical model trained over sequences of words to learn a probability distribution for next-word prediction. This inherent difference in how those features are generated, jointly with the classical debate separating statistical and syntactical features, is what drives such grouping. It is, however, our view that this is arbitrary from the brain's point of view, and we also point towards this false dichotomy in our main text. The analysis of each individual feature and their link to putative distinct cognitive processes though is beyond the scope of our analysis. We only focus on the distinction between them for the phase-amplitude coupling analysis which is our main result. The scoring partition shown in Figure 3 is mostly motivating the use of a **combination** of all those features. Although it is true that there is an interest in showing the contribution of individual features in a similar fashion as in Figure 3 panel c, we opted to add this analysis as a Supplementary Figure for the sake of completeness (supplementary figure 2, which we also mentioned in the main text).*

6. **The inconsistency in model comparisons further impacts the clarity of the interpretations in the discussion. Specifically, the claim (in the discussion) that different features (semantic/syntactic) have distinct effects on the oscillatory patterns/frequency bands. This statement is critical for the discussion but seems not so be fully supported either by the ITPC or PAC results for individual models. For ITPC (fig. 3) it is not clear if the reconstruction scores or the model-fit (r) can distinguish between different semantic and syntactic features. For PAC the results seem clearer (Fig.4,5,6) but it is not clear that statistical/semantic features are preferentially modulating theta-gamma coupling and syntactic by delta-beta. For instance, robust semantic/statistical Surprisal effects are driving delta-beta**

coupling, and the effects seems more extended across lags and source space than either of the Depth or Close models. Where the spatiotemporal courses for individual feature reconstitutions (Fig. 5,6.) compared directly (e.g., spatiotemporal cluster permutation)? It would also be important to show that linguistic-features based model remain significant after modelling out the effects of acoustic envelope (see previous point). In the last section of the discussion authors acknowledge possible contribution of the envelope to low-frequency effects and argue for linguistic feature-related PAC effects and this interpretation will be strengthened by including the envelope into the models.

We thank the Reviewer for pointing out those inconsistencies and our impreciseness. We rewrote the part of the Discussion which focused on those effects, as they were indeed too loosely related to the results observed. There is no clear separation between rule-based and statistical features in terms of score improvement over the base model. Both are contributing (which is the point being made). However, we do observe a significant difference between both sets, only in the delta band, and now report it in the discussion as (l. 302, Discussion):

We found that both feature sets consistently drive an increase relative to our base model (envelope and word onset) across frequency bands (see Figure 3). Only in the delta band we observed a significant improvement of Rule-based feature over Statistical ones.

Your point is also valid concerning the PAC results, but we are not associating theta-gamma or delta-beta to either feature group anymore. Instead, we now discuss the impact of individual features in light of the potential cognitive process they relate to.

Concerning the statistics, we did carry out a spatiotemporal cluster permutation to show the spatial coverage of feature-dependent PAC. This is now explained in the text in both the Results and the Methods sections.

Finally, we added the acoustic envelope to the base model (which now contains both word onset and envelope). Every model comparison which mentions a score improvement is done against this base model, and every comparison of TRF coefficients (so the PAC results of Figure 5, bottom panels) is made between a model containing all features (including acoustic envelope) and a model where only the feature of interest (for which we want to assess the significance of spatiotemporal coefficient values) has been randomized.

Minor:

1. There is something odd with Figure 2 – for me the power density (top plot, panels A B) and spectral modulation (plot G) appear empty hence it is not possible to evaluate those features of the MEG signal. I am not sure if something went wrong when uploading/formatting of that figure.

Thank you for bringing this to our attention. We have redesigned all figures, fixing font size, adding labels, and making sure all panels are visible, and believe that this issue is now resolved .

2. Figures overall need some cleaning up: please make sure the panel names (A,B,C) appear consistently in each figure and are similarly formatted. Some figures are missing axis labels (please double check) and label fonts are too small for some of the

plots. The terminology for model names is not consistent (Rule-base/Syntax; Stats/Statistical).

Thank you for your comments. We have now ensured that the panel names appear consistently in each figure and are similarly formatted. We have also ensured that all figures have axis labels and that the label fonts are consistent. The terminology for model names is now consistently called Rule-based and Statistical.

Reviewer #3

This study explores an interesting and important question in a rapidly raising domain of research. The authors used TRF and PAC analyses to study the neural encoding of word surprisal and sentence structure. The idea is generally interesting. It is indeed revisiting concepts that have been explored quite a bit in the literature, but I am not aware of other studies explicitly looking into these specific elements directly and simultaneously. So, I am of the opinion that the study is novel. However, I found the key results not convincing in the current version of the manuscript. Part of the problem is the manuscript had several quality issues, from the narrative that was missing crucial elements, to the figure, which are not of a sufficient quality. This overall lack of attention to graphical details and explanations made the paper hard to read, potentially hiding some elements of interest. Please see my comments below. My opinion is that the implications of the proposed analysis will be useful in the corresponding domain of research, leading to many citations. Nonetheless, I consider this result fairly technical, and possibly not of interest to the broader readership of NCOMMS.

Major comments

- 1. The first major issue is methodological. I did not understand how the authors can claim that the effects they find are not due to the acoustics of the stimuli. Is that something I missed? Otherwise, the authors should run further analyses to verify that this is not the case (e.g., I suggested one way to do so in my comment about the TRF analysis below)**

As suggested, we ran a new analysis to eliminate the possibility of an effect due to the acoustics of the stimulus. In the new pipeline, there are two types of controls which help to exclude the acoustics as a source of the observed effect. First, we added the envelope as a feature in all TRF models; this will capture variance in neural signals that are mainly correlated with the acoustic envelope. We then show the improvement compared to a model with only acoustic and word onset for both score and TRF (PAC) coefficients. Finally, we also did another analysis computing the coefficient for PAC for the French condition, for which there is, to some extent, acoustics, phonetics, and prosodic structure being processed but no language comprehension. There, we found an absence of PAC in relation to linguistic features (Supplementary Figure 4, to be compared with SI fig.3 or main text figure 5).

- 2. The second issue is the overall quality of the writing and figures. The narrative is jumping between sections with little explanations. The experiment is only explained in the Online Methods. Certain terms are defined too late and not used (e.g., TRF). The figures lack detail (as well as graphical attention e.g., font-size, missing labels). Statistical analyses are often only vaguely described.**

We thank the Reviewer for their fair attention to important missing or overlooked details. We have rewritten many parts of the Results, notably to add explanations of the experimental design and analysis pipeline. We made sure every abbreviation was defined early on. We

rewrote the Introduction and hope the Reviewer will find it more streamlined and clearer. All figures have been reproduced from scratch, correcting for font type and size, adding labels and enriching captions with explanations of the statistics, which are also now better reported in the main text. We hope the Reviewers will appreciate those changes and agree that the overall quality of our manuscript has gained from our revisions.

Other comments

1. Line numbers are missing

Indeed, we apologize for the oversight. Line numbers are now present in the revised manuscript.

2. Introduction: “However, language comprehension [...]”. That sentence is unclear. What does “not fundamentally predictable” mean exactly, and how is that consistent with the text in the parenthesis?

Indeed, this sentence was unclear. We meant that there is a fundamental aspect of speech or language that is unpredictable. Namely, the message to be decoded is novel, this is the information transmitted to the listener. However, some information may be redundant and predictable from, e.g., the structure of sentences. We have rephrased the sentence to make it clearer. The revised sentence reads as follows (line 10-13, Introduction, 2nd paragraph):

Although language comprehension is remarkably adaptable to varied inputs, the inherent unpredictability of novel messages—each encoding a unique intended meaning from the speaker—poses a challenge. This unpredictability is not absolute but relates to the relevance and specificity of each message, which often contains sequences and structures that are nevertheless anticipated based on knowledge from previous linguistic experience.

3. Last paragraph on that same page: It appears that only one research question is formulated, as the second point appears to be an operationalisation of the first question (i.e., how one would actually test the first question).

We thank the reviewer for pointing this out, it was indeed not a second question per se but rather an added detail over, or rephrasing of, the first question. Paragraphs of the introduction have been rewritten and rearranged. In the revised manuscript, we formulate the research question as follows (l. 51-58):

*In light of this apparent dichotomy, and in the context of the debate in cognitive science regarding the role of statistical information in language processing [42-46], we then ask *to what extent do their individual contributions explain neuroimaging data, is the whole better than the sum of its parts?* We hypothesise that they jointly contribute to explaining variance in the MEG data while presenting overlapping spatio-temporal sources. Moreover, the dynamics might disentangle them further as predictions and statistical inference seem to be a widespread phenomenon in cortical computation, while the organisation of linguistic units into nested hierarchical structures, at least at first blush, may be related to hierarchical processing in other domains in some ways, but not others [47-52]. We thus further hypothesise that brain responses to statistical and structural features are operated synchronously, with potential distinct time scales, and orchestrated through cross-frequency coupling.*

4. **Introduction, page 2, paragraph 1: It is unclear to me whether this hypothesis is novel (maybe a few references in that part would help). And it is unclear how this hypothesis is derived exactly. Is this inspired from the large literature on syntactic and semantic violations?**

We agree that this paragraph was unclear. Our hypothesis, however, is not inspired directly from the literature on syntactic and semantic violations. In our new introduction, we give more details about the literature inspiring this work (e.g. in the penultimate and the antepenultimate paragraphs). For instance, we now introduce the hypothesis with (l. 85-97):

Our hypothesis draws on the intersection of syntactic processing and predictive coding theories, positing that the brain's response to language is not just reactive but anticipatory, integrating both structural and statistical cues in real-time. We thus propose to link properties of syntactic structures, jointly with information-theoretic metrics to MEG data. Brennan and Pylkkänen [80] and Nelson *et al.* [26] have used a similar approach to link syntactic features to electrophysiological data. However, the former studies focused on the localisation of MEG activity to study word-evoked responses, while the latter analysed high gamma activity recorded from intracranial electrodes. In another study, Brennan & Hale [81] used information theoretic metrics built from context-free grammar parsers and delexicalised n-grams, which do not capture semantic information (thus their surprisal greatly differ from ours). Finally, in [22], a link is made between hierarchical syntactic features (node count) and surprisal from Markov models (n-grams, lexicalised and unlexicalised). While they elegantly show how different parsing strategies affect the prediction of neural activity, the statistics of word predictions are overlooked. Modern autoregressive language models produce a more precise estimate of conditional probabilities, from which we extract surprisal and also entropy, thus allowing for a better account of predictive mechanisms. To our knowledge, the current literature has not explored the interplay of such features across frequency bands with MEG.

5. **Paragraph 2: This part is vague and it makes it sound as if the results mentioned in the references were all talking about “cortical oscillations”, which is not the case. A bit more structure would help this paragraph separating evidence on the relevance of certain EEG/MEG frequencies vs. the hypotheses on what “low-frequency [cortical] oscillations” reflect. (I note that this issue is appropriately mentioned later on in the introduction).**

We thank the Reviewer for raising this inconsistency. The discussion about putative “oscillations” is now grouped in that same paragraph, and we removed the term “oscillations” when we were speaking about the band-limited activity (e.g. to refer to results reported for low-frequency cortical signals, as in the theta or delta band). We hope that the new text is clearer.

6. **Results: First paragraph. I suggest replacing this sentence with a brief description of the analysis pipeline. I did not like that the paper jumps into the results without explaining the experiment.**

Thanks for bringing this to our attention, as the Methods were originally not written to be after the Results we overlooked the importance of laying out the basis of the pipeline once

more for this section. We have now completely rewritten the entry to the Results section, which now reads (l.116-129, Results, 1st paragraph):

In order to measure time-resolved cross-frequency coupling in relationship to different linguistic features we first analyse both the presence of word-related phase and power modulation as well as how speech representations perform in predicting MEG signals. First, we extracted linguistic features reflecting both syntactic complexity and statistical properties of speech from the presented stimuli. So called rule-based features are derived from constituency tree, while statistical features are generated using the probability distribution of next word predictions from a large language model. These features include syntactic depth and the number of closing brackets to represent structural properties of syntax [29,80], and word surprisal and entropy for statistics of word-level predictions [26,40]. We construct Temporal Response Function (TRF) models to predict MEG signals based on these linguistic features, enabling a precise examination of how each feature influences neural responses over time. The analysis pipeline (detailed in the Methods section) involves aligning the MEG data with our linguistic features, then using ridge regression to estimate the TRF coefficients, and finally evaluating the models' performance through correlation analysis between predicted and observed neural activities. In most cases, model comparison is carried out between the true model and a null model for which values (but not timing) of a given feature (or of a feature set) are shuffled. Through this method, we aim to reveal the mechanisms by which the brain integrates and processes linguistic information at different levels. The general analysis pipeline, along with stimulus representations and analysis methods, are presented in the diagram of figure 1.

- 7. Second paragraph. I think the claim that PSD is studied to assess the “presence of the expected neural oscillations” goes counter to the beautiful introductory paragraph on oscillations vs. evoked responses. I suggest adding a sentence to the introduction clarifying what the term “neural oscillations” refers to exactly. In my view (at this point of the reading), the authors are talking about the MEG power within a specific frequency-bin, rather than “neural” oscillations of an endogenous nature. A simple clarification of the specific use of “neural oscillations” would suffice. Or possibly changing that term to band-limited MEG power, or something similar.**

We thank the Reviewer for bringing this crucial point to light. We decided to clarify our terminology in the introduction to refer to high-frequency band-limited power “peaks” as oscillations. However, throughout the manuscript, we also chose to refer to such activity unambiguously as neural activity within specific frequency bands. In the new introduction, we now wrote (l.82-84, Introduction):

While we prefer to speak about low- and high-frequency activity unambiguously, we may use the term cortical oscillations to refer to band-limited power elevation observed in the MEG power spectra without making any claim about the underlying mechanism.

- 8. A few comments on Figure 2: I don't see the letters indicating the panels.. also, the mid-left panel is not showing the information appropriately, as the red lines are almost entirely hidden (maybe use transparency?). The y-label is missing. Mid-right panel: please rotate delta and theta. Also, the legends should include a name and legend. Are 4 different scales really necessary? Font-size seems different across different panels. Mid panels: it's not entirely clear to me what those horizontal solid lines actually indicate. I got the point in the end (top row delta and**

bottom row theta), but do we really need such confusing lines. Instead, I suggest indicating delta and theta on the mid-left plot. Does the figure only show the actual values, or does any of those results include the result of the statistical significance test (a part from the top-left panel, which however is not explained appropriately in the caption – what is the yellow area exactly? $p < 0.05$? with what test?)

We have now addressed all those comments in the new figure. Panel letters are shown, we did not change the transparency as it did not really solve the issue (lines are overlapping, which is to say that there is no strong difference between power in either condition), y-label has been added, and we rotated theta and delta labels, font sizes are the same now across all figures (and panels), we removed the line clutter and indicated delta and theta as suggested and we explain the statistical testing directly in the caption too. The caption now reads:

Sanity checks, showing power spectral density (PSD) of the MEG data, the stimuli and word-related phase and power modulation. a: and b: PSD of MEG averaged over sensors for French and Dutch listening conditions. A cluster-based permutation test revealed no significant difference between average power, although the permutation revealed a marginally significant cluster in the beta band, showed with the shaded area (p -value = 0.06). The topographic inset presents the power difference within this frequency band, marked sensor are sensors for which the difference were significant (paired t-test with $\alpha = 0.05$, dof=24, fdr-corrected). b: PSD of the acoustic envelope average across stories within each condition, no significant difference found (using cluster-based permutation using independent t-tests as cluster statistics). c: Cerebro-acoustic coherence. We computed the magnitude squared coherence between MEG sensor data and speech envelopes. The shaded areas are clusters with a cluster p -value below 0.05. Note that the coherence is actually greater for the French condition. The first column of topographic plots on the right indicate average coherence values in the given regions; in the second column we show the difference contrast between Dutch and French conditions within those frequency bands. d and e: Inter-trial (time-locked on word onsets) phase clustering (ITPC) and power modulation respectively, averaged across sensors. The contour outlines significant time-frequency cluster (cluster-based permutation using one sample t-test on data after baseline removal, therefore testing for difference w.r.t baseline). Using a similar statistical approach, we did not find any significant difference between French and Dutch listening conditions.

- 9. Overall, the second paragraph of the results is somewhat vague. More details clarifying how those results were actually obtained should be included. I appreciate the attempt of going straight to the point, but the authors should provide all the essential details to the reader at this point (especially on what exactly is tested in the statistical analysis). So, in sum, this paragraph should be clearer (e.g., the result for the mid-panels is unclear – where is the result of the cluster statistics shown exactly? Is it in the right panel? So, are those values thresholded by that statistical analysis?)**

We agree with the reviewer that the results section was neither well-structured nor detailed enough for the reader to apprehend the figures presented. We rewrote the introduction to the results (first two paragraphs) to give more details on the method and analysis used. Moreover, each figure is better explained with captions that mention what statistical analysis has been done (and in particular, if there is thresholding or not, at which value, and how the p -values

were corrected) and with added details in the main text. We hope the reviewer will find our amendments have improved the manuscript in that respect.

- 10. Figure 3: The authors should add an explanation of “depth close” earlier in the manuscript. Cosmetic comments: the font-size is absolutely inconsistent across the figure. Y-labels are missing. The meaning of the shading in A and B is not entirely clear to me. Panel D: datapoints appears to be hidden.**

We have now added a short description of the “depth” and “close” features in the Introduction, and a slightly more detailed one in the Results section (on top of the explanations found in the methods, which indeed appear later). Those sentences read (l.30-32, Introduction):

These syntactic metrics have been used to study the effect of syntactic operations such as unification (the “merge” operation in the minimalist program [15,25]) or integration of an item into a larger structure and the depth, a proxy for ongoing complexity, of the syntactic tree at a given word [26-28].

And (l.175-178, Results: Joint Contributions...):

*Namely, we built two rule-based features, both derived from syntactic constituency trees: *close* designates the number of phrasal constituents a given word is closing (thus counting closing brackets at each word), while *depth* stands as a proxy of ongoing syntactic complexity simply by measuring how deep in the hierarchy a given word is.*

We have adapted the font size of every figure and made sure every axis is correctly labelled. Finally, the shaded area, which corresponds to a region highlighted by the cluster-permutation test, is now explained directly in the caption.

- 11. Figure 5: fix overlap of text on top-left. The authors seem to use yet another style for indicating the panels (lowercase, on the bottom of the panel). The figure caption is less than minimal. All the necessary details should added instead (e.g., what do the shaded areas indicate exactly?). Again x- and y-labels are missing!**

All figures have been completely redone, respecting guidelines for font and panel labelling, we hope this is now resolved.

- 12. Section 2.1: Statistical results should be clarified and reported. Also, please refer to the specific panels when mentioning a figure.**

We now report all statistical analysis in the results section and rewrote the paragraph concerning statistics in the Methods section too. Panel labels are now added to every figure and correctly referenced in the main text.

- 13. I have a more fundamental question about section 2.1. Did the authors check that the increase is due to the actual information in the features rather than the larger number of parameters (e.g., by using shuffled features, rather than subsets of features)?**

This is a very important point indeed. We indeed computed the improvement in reconstruction accuracy against null models (so features were shuffled rather than removed). However, we were not doing so consistently. We have now reran all model comparisons, by respecting this principle to control for the number of regressors. This is now explained in several instances throughout the text (relevant Results sections and in Methods, as well as some captions). We added a sentence right at the beginning of the results section to clarify this:

Line 126, Results:

In most cases, model comparison is carried out between the true model and a null model for which values (but not timing) of a given feature (or of a feature set) are shuffled.

Line 183-189, Results: Joint contribution...

Every model is then matched in a number of regressors by swapping the feature of interest with a null feature, which follows the same statistics. Each null model for a given feature set consists of a TRF computed by using a shuffled version of the stimulus features (the shuffling is strict, as we keep word onset intact and shift values of linguistic features by several words). For instance, to compute the relative increase in reconstruction accuracy of Rule-based features (as seen in figure 3, panel **d), we alter the values of the *close* and *depth* features while keeping their original timings. By comparing the score to baseline null models, we normalise for the increasing number of features as each null model contains the exact same number of regressors.**

Moreover, we also updated the caption to reflect this point.

- 14. Section 2.1: The text should clarify that a temporal response functions is estimated here with a lagged regression, or maybe deconvolution could be mentioned. There are many ways to explain this, but the current text (i.e., “linear forward model”) is insufficient. Also, the methods section should clarify which exact implementation is used (is it Eelbrain, the mTRF-Toolbox, or maybe something custom?)**

Thanks for pointing this out; we have corrected the sentence, which now reads (l.180-181, Results):

We first analysed the spatio-temporal dynamics of low-frequency activity in response to all features using a convolutional model which consists in estimating so-called temporal response functions (TRFs).

All the code to compute TRFs was custom-based (and made available; the current code is built on top of the Python code previously shared by Weissbart et al., 2020).

- 15. From Figure 1, I gather that no acoustic feature was included in the model e.g., envelope, spectrogram. Why is that the case?**

Indeed, our previous analysis used only word-level features and as such, our word onset features were the lowest level of control. We had in mind to keep only features of the same type (as it can happen that the regularisation required by a continuous feature, such as the

acoustic envelope, is too much for other impulse-like features, leading to underfitting for those other regressors). However, we now incorporated the acoustic envelope, as the regularisation it requires does not impede the estimation of other coefficients as far as we can tell. Therefore, we now include two acoustic controls: word onset to “collect” all word-level effects that are not explained by other linguistic features, and the acoustic envelope. Now, our Base model (also better described in the new Method sections, which now shows a summary table for each model, table 1) thus contains null/shuffled versions for all linguistic features plus the intact acoustic envelope and word onsets.

- 16. A related question: Could the increase mentioned in section 2.1 be due to acoustic differences captured by the “syntactic features”? This should be verified either by studying the stimulus itself and/or by considering the previous comment.**

Please see our response to 15 above. Note that we also present another control in support of our result, computing the PAC per feature using the TRF method on the French (uncomprehended) stories, shown in SI figure 4.

- 17. Section 2.3. The first sentence is empty. The only way to know what that means is to look at those references. Could the authors be more direct?**

Indeed, the sentence did not convey any information besides the references. We changed it to the following couple of sentences (l.144-152, Results: Phase consistency and ...):

*Recent studies have demonstrated the important role of low-frequency cortical activity during speech processing. Whether it is a form of neural entrainment to the acoustic envelope or a series of evoked responses to acoustic edges is debated, but results come together in that they show a stronger correlation between speech spectro-temporal features and MEG or EEG signals in both the delta and theta band [53,56,67]. Etard & Reichenbach [84], for instance, concluded from their study the existence of a dissociable account for each frequency band with respect to the clarity and intelligibility of the speech signal. Modelling work from Hyafil *et al.* [59] reinforces the idea of a phase alignment within the theta band to promote segmentation of syllabic sequences. Altogether, we hypothesised change in phase consistency in delta and theta range after word onset, potentially coupled with modulation of beta oscillatory activity and with gamma broadband activity (which has also been suggested by [65,85]).*

- 18. Methods. It seems that unnecessary filters were applied (for example, a low-pass and then a band-pass i.e., in that case, a low-pass and a high-pass would have been sufficient), which introduces additional unwanted artifacts that can be problematic (de Cheveigne and Nelken, Neuron, 2019).**

We may have overlooked this issue in our previous analysis. However, we are indeed low-passing the signals once before resampling the raw data to avoid aliasing. This is a filter designed at 1.2kHz. We then work on the resampled signal for computational and memory efficiency. However, we only band-passed data once after this initial resampling (so one high-pass filter and one low-pass, designed at 200Hz). While we could have initially band passed and then resampled, we opted to apply the band pass filter at the new (lower) sampling frequency as the design of a band pass filter at 1.2kHz with low corner frequencies can result in unstable filter and/or longer filter length.

Reviewer #1 (Remarks to the Author):

The authors have put a considerable amount effort into this revision and produced a clearly improved paper. Results are interesting and their significance will be high in the field and beyond.

Reviewer #2 (Remarks to the Author):

Authors have made all requested edits both to the analyses and the manuscript since my original review. The analyses are sound, appropriately detailed and their interpretation is clear and consistent. The methods and results are novel and of significant interest to scientific community. I would recommend this for publication.

Reviewer #3 (Remarks to the Author):

The authors did a great job in addressing my comments. There is just a minor comment left (please see below). In my view, the study is well done and interesting. Well done!

- Figure 1: the figure caption does not seem to match the figure now i.e., where are the letters E, F, and G? Also, use the same capitalisation.

Reviewer #3 (Remarks on code availability):

That's fantastic work, well done. One thing that it would be nice to have (and that I have seen before in NCOMMS) is code to generate the exact same plots in the paper. While I have only code for part of the figures in the paper (but maybe I missed that?). Again, that's something that I'll leave to the editor. I think it would be useful, but that might not be a requirement for NCOMMS.

Point-by-point response to reviewers

Reviewers' comments are shown verbatim in normal font weight; our response is in *italics*.

Reviewers 1 and 2 did request further edits and complimented the work, to which we are grateful. Thus, we are only presenting reviewer 3's comment here.

Reviewer 3

The authors did a great job in addressing my comments. There is just a minor comment left (please see below). In my view, the study is well done and interesting. Well done!

- Figure 1: the figure caption does not seem to match the figure now i.e., where are the letters E, F, and G? Also, use the same capitalisation.

> Indeed, while revisiting the figure and improving on it, we did not add the caption for the new panels. This is now fixed.

Remarks on code availability

That's fantastic work, well done. One thing that it would be nice to have (and that I have seen before in NCOMMS) is code to generate the exact same plots in the paper. While I have only code for part of the figures in the paper (but maybe I missed that?). Again, that's something that I'll leave to the editor. I think it would be useful, but that might not be a requirement for NCOMMS.

> This is indeed an excellent suggestion, and we now have added a script in the repository for the code allowing to reproduce all figures using the preprocessed and source data deposited in Figshare.